# Pheomelanin pigment remnants mapped in fossils of an extinct mammal

Phillip L. Manning [1,2], Nicholas P. Edwards[3], Uwe Bergmann [4], Jennifer Anné[5], William I. Sellers [1], Arjen van Veelen[6], Dimosthenis Sokaras[3], Victoria M. Egerton[1,5], Roberto Alonso-Mori [7], Konstantin Ignatyev [8], Bart E. van Dongen[1], Kazumasa Wakamatsu [9], Shosuke Ito[9], Fabien Knoll [1,10] & Roy A. Wogelius[11]

Recent progress has been made in paleontology with respect to resolving pigmentation in fossil material. Morphological identification of fossilized melanosomes has been one approach, while a second methodology using chemical imaging and spectroscopy has also provided critical information particularly concerning eumelanin (black pigment) residue. In this work we develop the chemical imaging methodology to show that organosulfur-Zn complexes are indicators of pheomelanin (red pigment) in extant and fossil soft tissue and that the mapping of these residual biochemical compounds can be used to restore melanin pigment distribution in a 3 million year old extinct mammal species (*Apodemus atavus*). Synchotron Rapid Scanning X-ray Fluorescence imaging showed that the distributions of Zn and organic S are correlated within this fossil fur just as in pheomelanin-rich modern integument. Furthermore, Zn coordination chemistry within this fossil fur is closely comparable to that determined from pheomelanin-rich fur and hair standards. The non-destructive methods presented here provide a protocol for detecting residual pheomelanin in precious specimens.

[1] University of Manchester, School of Earth and Environmental Sciences, Interdisciplinary Centre for Ancient Life, Manchester M13 9PL, UK. [2] Department of Geology and Environmental Geoscience, College of Charleston, 66 George St, Charleston, SC 29424, USA. [3] Stanford Synchrotron Radiation Lightsource, SLAC National Accelerator Laboratory, Menlo Park, CA 94025, USA. [4] Stanford PULSE Institute, SLAC National Accelerator Laboratory, Menlo Park, CA 94025, USA. [5] The Children's Museum of Indianpolis, 3000 N Meridian St, Indianapolis, IN 46208, USA. [6] University of Southampton, Faculty of Engineering and Physical Sciences, Southampton SO17 1BJ, UK. [7] Linac Coherent Light Source, SLAC National Accelerator Laboratory, Menlo Park, CA 94025, USA. [8] Diamond Light Source, Didcot OX11 0DE, UK. [9] Department of Chemistry, Fujita Health University School of Health Sciences, Toyoake, Aichi 470-1192, Japan. [10] ARAID—Fundación Conjunto Paleontológico de Teruel-Dinópolis, 44002 Teruel, Spain. [11] University of Manchester, School of Earth and Environmental Sciences, Williamson Research Centre for Molecular Environmental Science & Interdisciplinary Centre for Ancient Life, Manchester M13 9PL, UK. Correspondence and requests for materials should be addressed to R.A.W. (email: roy.wogelius@manchester.ac.uk)

One of the main challenges of paleontology is to relate the residues preserved in fossils to understanding of the lifeform itself. In this regard, soft tissue residue is notoriously difficult to study. Synchrotron rapid scanning-X-ray fluorescence (SRS-XRF) imaging combined with X-ray absorption spectroscopy (XAS) has provided new information concerning biomarkers for pigment and other components of animal and plant tissue[1–15]. Integumentary pigmentation is of particular interest, as it plays key physiological and behavioral roles and can change relatively quickly on evolutionary timescales with important consequences for the species. Melanin is one of the critical pigments governing color in ancient and modern lifeforms. There are two main melanin pigments in animal tissues, eumelanin and pheomelanin. The former is more prevalent (>75%) and produces dark black or brown hues in both invertebrates and vertebrates[16]. Pheomelanins, in contrast, form the lighter reddish-brown pigments. Trace metals are key components of melanin and play important roles in melanogenesis. Melanins are complex molecules formed from the aromatic amino acid tyrosine[17] via the action of the Cu-containing enzyme tyrosinase. Because Cu is the metal cofactor in the enzymatic process forming eumelanin[18], elevated concentrations of organically bound Cu can typically be correlated with eumelanin-rich tissue[2,19,20]. After Cu, Zn is the second most abundant metal in mammal melanosomes[21]. Both metals may be complexed within the interior ring structure of eumelanin[2], attached to the diol functional group of dihydroxyindoles, or attached to terminal carboxylate groups, but in all cases, the Cu and Zn within eumelanin are strictly light-element coordinated (O/N): in eumelanin, there are no sulfur groups to which trace metals can bind. Importantly for this study, pheomelanin synthesis additionally requires the sulfur containing amino acid cysteine as a substrate. Sulfur in pheomelanin is contained within benzothiazole (or benzothiazine) units that are accessible for metal attachment. While Cu is strongly associated with eumelanin, Zn correlates with pheomelanin pigment[1]. Previously, work on fossil integument concluded that organosulfur–Zn complexes may be the residue of pheomelanin[7]; however, detailed coordination chemistry for Zn in extant pheomelanin to use in comparison with the fossils was unavailable. Subsequently, Edwards et al.[1] applied detailed XRF and XAS to extant pheomelanin-rich feathers. The results showed that these feathers possessed a distinct chemical signature for Zn and S, with a significant portion of the Zn inventory bonded to S, almost certainly through the S contained within the pheomelanin molecule. This conclusion was based on the fact that there was a strong, spatially resolvable correlation between Zn and pheomelanin-associated sulfur groups. In light of these new results with extant integument, the work presented here tests whether pheomelanin residue can now be resolved and mapped in extinct organisms. (Further information on melanin is contained within Supplementary Note 1: Melanin Chemistry—background.)

Multiple studies have comprehensively shown that fossil eumelanosomes can contain several different biomarkers, including organic–metal chelates[2,20] and specific molecular weight organic fragment patterns[20,22,23], that are consistent with eumelanin pigment. In summary, the first synchrotron-based study used XAS to detect organic complexes of Cu in fossils that are directly comparable with eumelanin–Cu complexes in extant organisms[2]. SRS-XRF was then used to map whole-organism Cu distribution in order to resolve pigment patterns in an important extinct avian species (*Confuciusornis sanctus*, ~120 Mya). Glass et al.[22] subsequently showed the chemical affinity between a modern *Sepia officinalis* eumelanin standard and organic residue within >160 Mya fossilized cephalopod ink sacs and showed that there is a clear chemical difference between the fossil ink sacs and

the embedding sedimentary matrix. Those findings relied on perhaps the best diagnostic method for resolving melanin in soft tissue: alkaline hydrogen peroxide oxidation pretreatment with subsequent high-performance liquid chromatography analysis (AHPO-HPLC) of the product[24]. Colleary et al.[23] set out to compare morphology with chemistry in a wide set of specimens, including experimentally reacted standards, in order to address a key controversy in fossil pigment research: previous work that had relied on a microscopic structure had been challenged due to the ambiguity of using microbody morphology as a diagnostic indicator of the presence of fossil eumelanosomes or pheomelanosomes, because mineralized microbes could potentially mimic the structure of these organelles[25]. Time-of-flight-secondary ion mass spectrometry (ToF-SIMS) was used as a qualitative method for detecting molecular fragments of melanin or degraded melanin residue. Colleary et al.[23] had several important conclusions: (1) the putative eumelanosomal microbodies have a clear chemical affinity with eumelanin and hence are more likely to be preserved organelles than microbial mats, (2) bacterial biomarkers are not correlated with the proposed melanized fossil regions, and so again a microbial origin is not supported chemically, (3) chemical residue may persist even though melanosomal structures are destroyed, and (4) pheomelanin markers were not resolved via ToF-SIMS for any of the fossils examined, despite the fact that melanized soft tissue in the animal kingdom typically contains both pigments[26]. This final point is crucial with respect to detecting pheomelanin residue. Their principal component analysis strongly implies that the collapse in their second principal component (PC2) as a function of age (or degradation) is most likely a direct function of sulfur loss, as the sulfurous smell of their experimentally reacted pheomelanin corroborates. Pure eumelanin does not contain sulfur, and so the documented collapse in PC2 in their work most likely explains why the pheomelanin biomarkers are not detectable in the ToF-SIMS analyses: the pheomelanin biomarkers become undetectable relative to the eumelanin biomarkers using their method, as sulfur compounds are devolatilized from the organic residue. Finally, the most recent advanced chemical study completed a comprehensive chemical analysis of an ~55 Mya fossil feather using a range of methods, including ToF-SIMS and XAS[20]. They also found that the microbodies they resolved via SEM contained biomarkers for eumelanin, as determined by ToF-SIMS, and presented microscopic (0.02 mm²) ToF-SIMS maps of a eumelanized region. Their chemical analysis showed organic sulfur to be present, but this was ascribed to diagenetic addition of sulfur rather than pheomelanin residue, despite the presence of pheomelanin-related peaks.

As discussed above, we have previously shown how synchrotron imaging and spectroscopy can resolve pheomelanin in extant soft tissue and proven the validity of our approach via direct comparison to AHPO-HPLC analyses of the same specimens[1]. However, as also discussed above, the problem of resolving biomarkers for pheomelanin in fossil material has not yet been solved. This is problematic, given the usual co-occurrence of eumelanin and pheomelanin[1]: even dark black human hair includes, on average, 15% pheomelanin[26]. We also note that, in terms of methodology, while AHPO-HPLC is the best quantitative diagnostic tool, it is a destructive nonspatial technique, and therefore cannot provide information on patterning and cannot be employed on extremely rare paleontological or archeological specimens. In contrast to AHPO-HPLC, ToF-SIMS is qualitative and is a microfocus technique. ToF-SIMS requires destructive subsampling and can only image microscopic areas (e.g., ~$10^{-4}$ cm²) of large organisms such as the avians, mammals, and fish (>$10^{2}$ cm²) that are of interest in vertebrate paleontology. Previous work has indeed shown that whole-organism extrapolations

from microscopic areas can be misleading[1,3]. Therefore, the work presented here seeks to test whether nondestructive and extremely sensitive synchrotron-based methods (including XAS) may be able to resolve traces of pheomelanin residue in fossil specimens and map this residue over the entire organism length scales, as we have shown is possible in extant tissue[1]. This approach may allow pigment distribution to be resolved even in severely altered specimens. In order to verify our results, we have completed additional analyses, including state-of-the-art destructive AHPO-HPLC measurements on standards, such that the synchrotron results with fossils may be directly compared with the pigment loadings in extant organisms.

Two phylogenetically well-constrained and taphonomically well-preserved fossil specimens of *Apodemus atavus*[27] from Willershausen (Germany) were selected, the holotype GZG. W.20027b (Figs. 1a, b and 2a–c) and a second specimen GZG. W.17393a (Fig. 2d–g). This Pliocene species of field mouse is closely related to the extant wide-ranging species *Apodemus sylvaticus* and *Apodemus flavicollis*[28]. Extant members of this genus are reddish colored and thus the phylogeny would imply that the related extinct species would also have contained significant amounts of pheomelanin pigment. (Further information concerning Willershausen is in Supplementary Note 2: Geological setting.)

Here, we show that traces of organic sulfur can be mapped and correlated with Zn within the integument of two different fossils of *A. atavus*. We also show that Zn is directly bonded to these associated organic sulfur compounds. Detailed comparison is made between the fossil chemistry and the chemistry of extant pigmented hair and fur to show that the coordination chemistry of Zn within the fossil is equivalent to the coordination chemistry of Zn within modern pheomelanin-pigmented tissue. Based on these findings, we conclude that the Zn–organosulfur compounds are the residue of pheomelanin and indicate that this species of mammal was pigmented with pheomelanin, similar to its extant-related species. Our results also help explain why the detection of pheomelanin pigment residue is challenging: the high original quantities of sulfur in keratinous integument produce degradation products rich in oxidized sulfur, which obscure the organic sulfur residue derived from a pigment. In order to resolve the pheomelanin signal embedded within the complex mixture of organic degradation products, the bonding environment for organically complexed Zn is apparently the most sensitive and most stable chemical indicator. Resolution of pheomelanin pigment residue should now be possible, using a combination of chemical imaging and X-ray spectroscopy, at least for specimens with an age equal to or less than 3 million years.

## Results

**Grazing-incidence XRD**. Grazing-incidence X-ray diffraction (XRD) analysis identified the minerals gypsum, calcite, quartz, ankerite, and muscovite in the sedimentary matrix plus hydroxyapatite in the fossil bone (Supplementary Fig. 1).

**SRS-XRF imaging**. The holotype of *A. atavus* (GZG.W.20027b) is preserved as an intact laterally compressed fossil (Fig. 1a, inset photograph of extant *A. sylvaticus* for comparison) and will hence be referred to as the lateral fossil. Figure 1b shows the corresponding false color Zn, P, and organic S composite SRS-XRF image. The organic S part of this three-component image was produced by tuning the monochromatic incident X-ray beam to a specific reduced sulfur oxidation state resonance (2472.5 eV), thereby excluding any contribution from oxidized inorganic sulfate to the image. This reveals that organic S and Zn are concentrated and correlated within the skin and parts of the fur:

high correlation is indicated by the bright yellow regions. Supplementary Fig. 2 presents computed correlation maps of Zn with organic S and Zn with Cu in both of the *A. atavus* specimens used in this study. The clear correlation of zinc with organic sulfur highlighted in Fig. 1b and Supplementary Fig. 2 robustly shows that the bulk of the Zn imaged is bound within the integument and is associated with organic S. (See the "Methods" section for details. Additional elemental maps are shown in Supplementary Fig. 3.)

Figure 2 provides further details of the S oxidation state mapping of the lateral fossil, and compares these results with the other, dorsoventrally compressed specimen of *A. atavus* (GZG. W.17393a, subsequently referred to as the dorsal fossil). Figure 2a presents the total sulfur inventory of the lateral fossil imaged at an incident beam energy of 3.150 keV, well above the critical excitation energy for all oxidation states of sulfur, including fully oxidized sulfate (~2.482 keV). High-sulfur concentrations in the surrounding matrix are due to the presence of sulfate contributed from the mineral gypsum ($CaSO_4 \cdot 2H_2O$: identified by grazing-incidence XRD on this specimen's bedding plane, see Supplementary Fig. 1). Figure 2b was obtained by mapping with the incident beam energy set below the sulfate absorption edge, at the energy equivalent to the absorption edge for an organic thiol sulfur species (2472.5 eV, calibrated using Zn–cysteine). Here, the inorganic bedding plane sulfate salts disappear, resulting in a higher contrast image of the remnant soft tissue. Furthermore, by using image subtraction (see Methods), the distribution of sulfate mineral precipitates on the bedding plane is isolated in Fig. 2c. These results are reproduced with the dorsal fossil specimen, and a direct comparison of Fig. 2a–c with d–f shows that exactly the same result is obtained from both specimens: sulfate is present on the bedding plane (Fig. 2f) and the reduced organic species of sulfur are concentrated in the soft tissue residue (Fig. 2e). Indeed, this allows us to spatially resolve the radiating texture of hair, along with patches of skin remnants (Fig. 2b, e). Figure 2g is a false-color composite image, which unequivocally shows that Zn and organic sulfur are correlated within the integument of the dorsal fossil just as in the lateral specimen. (Maps of the dorsal fossil are given in Supplementary Fig. 4a, b.)

**Sulfur X-ray spectroscopy**. Given that the fossil soft tissue displayed enrichment in organic sulfur compounds, and that both the structural protein (keratin) and red pigment (pheomelanin) in extant mouse fur contain sulfur species, we performed detailed sulfur X-ray absorption near-edge spectroscopy (XANES) in order to accurately characterize the sulfur inventory in the fossils. Figure 3 presents the results from fossil and extant material along with several sulfur standard compounds. All spectra are quite different from each other, reflecting the strong impact that changing the oxidation state and coordination environment has on XANES spectra for sulfur. Above the standard spectra, we present extant spectra for a dark red pigmented *Apodemus sylvaticus* and for an albino *Mus musculus*. Both are dominated by disulfide, due to alpha keratin being rich in disulfide moieties. The spectrum for the red-pigmented *A. sylvaticus* however clearly shows two features which are consistent with the presence of a sulfur-bearing heterocyclic compound such as benzothiazole: the appearance of the subtle small oscillation at ~2475.5 eV indicated by the red circle (and clearly shown in the magnified inset), plus the enhancement in the intensity of the second peak in the bifurcated maxima at ~2473.5, located at the vertical red-dashed line. A binary linear combination fit of benzothiazole plus disulfide for each extant specimen shows that the red fur contains ~9% (±2) heterocyclic sulfur and the albino fur contains ~6% (±2) heterocyclic sulfur. Organosulfur compounds have been

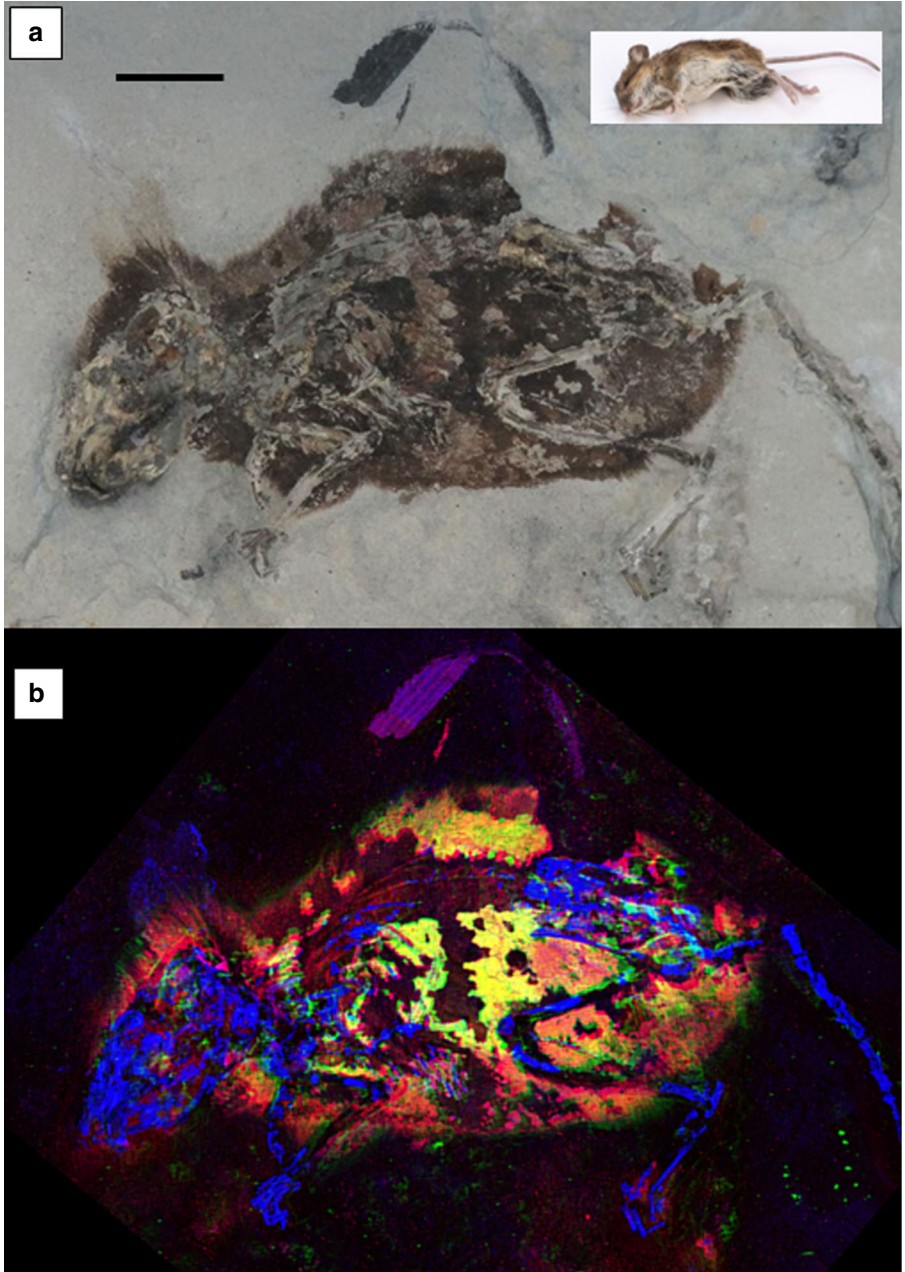

**Fig. 1** Optical and X-ray images of *Apodemus atavus* "lateral" fossil. **a** Optical image of "lateral" fossil *A. atavus* (GZG.W.20027b) with the inset of extant *A. sylvaticus* in the upper right for comparison (scale bars = 1 cm). **b** False-color SRS-XRF image reveals exceptional preservation of integument as well as bone. This image is a combination of three maps, two standard single-element maps (blue = P, green = Zn), plus a third map which has been produced to especially emphasize the distribution of a specific oxidation state of organic sulfur (red = S in thiol) in order to highlight the clear correlation between the distribution of Zn and organic sulfur which together appear as bright yellow. (Optical photograph by P.L.M.)

shown to be a key component of pheomelanized tissue and to bond directly to divalent metals[29]. Therefore, the heterocyclic sulfur in the red fur is almost certainly due to the presence of organosulfur compounds associated with pheomelanin pigment. The trace heterocyclic sulfur in the albino fur may be due to background levels of other sulfur compounds besides pheomelanin-related compounds, or may be caused by extremely low levels of pheomelanin, as documented for tyrosinase-positive types of albinism[30].

Five spectra from the lateral fossil surface are presented at the top of Fig. 3. Linear combination fitting (LCF) of the fossil spectra using all five reference spectra was performed in order to quantify the contribution of each sulfur oxidation state to each fossil spectrum (Supplementary Table 1). The sedimentary matrix, as expected, contains pure sulfate, a result corroborated by imaging and XRD analyses. Sulfur within the fossil bone is 97% sulfate, consistent with previous results for fossil bone[2,3,6,9,15]. The graminoid fragment (purple region at the top center of Fig. 1b) is mostly disulfide (24%) and sulfate (71%). In contrast, spectra obtained from two different locations within the integument on the lateral mouse fossil exhibit a significant quantity of a heterocyclic organic sulfur component, with benzothiazole-type groups contributing as much as 24% to the fit of the topmost spectrum. From these results, we may conclude that (1)

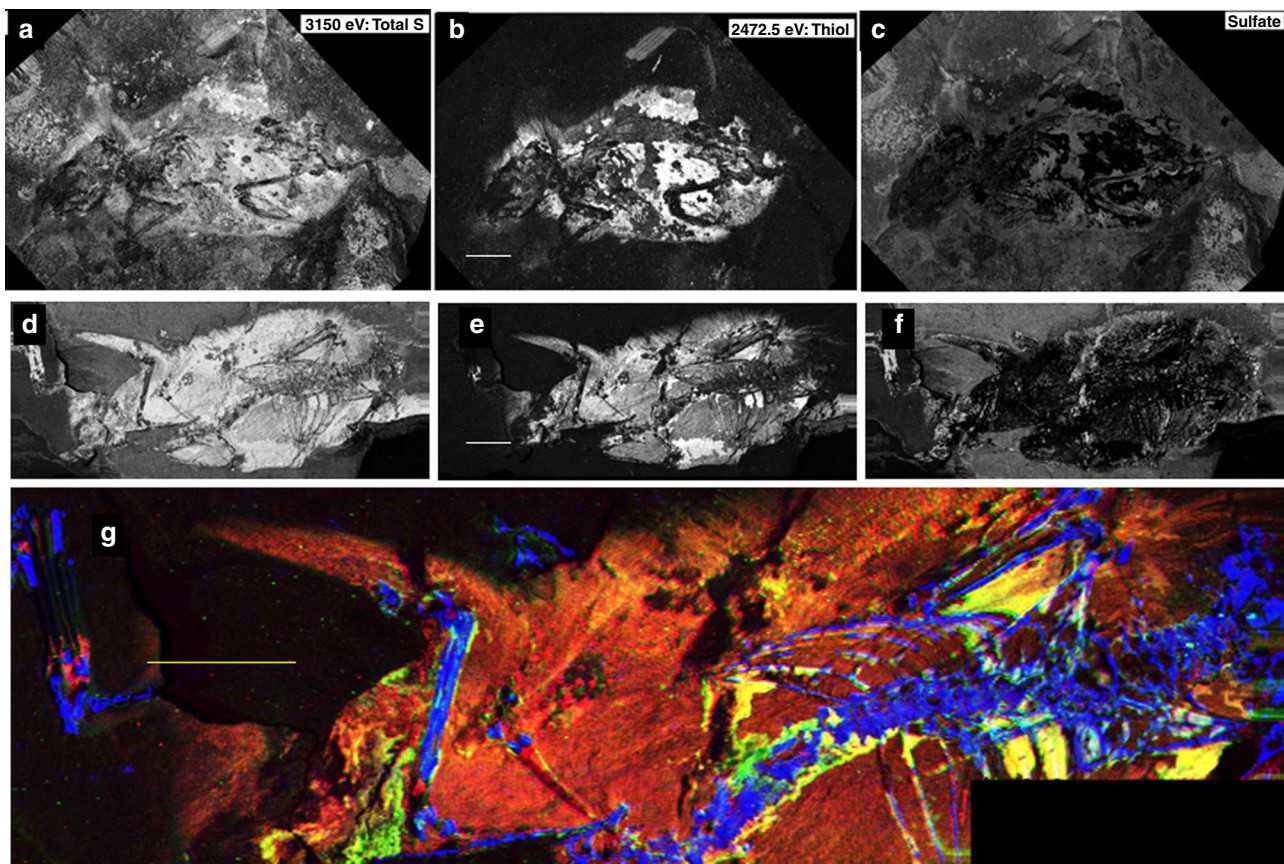

**Fig. 2** X-ray image comparison of "lateral" and "dorsal" fossils. SRS-XRF map comparisons for the *A. atavus* lateral fossil (GZG.W.20027b: **a–c**) with the "dorsal" fossil (GZG.W.17393a: **d–g**). Maps **a** and **d** are total sulfur (incident beam energy 3150 eV). **b** and **e** show reduced organic species of sulfur (incident beam energy 2472.5 eV). **c** and **f** subtract the organic image from the total sulfur map, thus showing only inorganic sulfate. **g** is a false-color map (blue = P, green = Zn, and red = organic S) for the dorsal fossil comparable to that shown in Fig. 1b for the lateral fossil (scale bars = 1 cm). Bright yellow areas indicate a correlation between Zn and organic sulfur

heterocyclic S is resolvable within the fossil soft tissue regions, and (2) these spectra document the breakdown of disulfide-rich alpha keratin and pheomelanin-derived benzothiazole to fully oxidized sulfate and reveal a distinct taphonomic reaction pathway that gives new insight into exceptional preservation. In particular, the presence and relative amounts of intermediate sulfur species in the fossil indicate a sulfur oxidation pathway that has only gone part way to completion. These expected intermediates on the well-known breakdown sequence of organic sulfur in soft tissue are consistent with the endogeneity of the compounds identified. No external source of sulfur is required to explain the chemistry detailed above. In fact, the quantities of sulfur in the fossil imply closed-system behavior with respect to sulfur, as the total quantities of sulfur are virtually identical in the fossil and in extant mouse fur (Supplementary Table 1). (Further information concerning the sulfur oxidation pathway is in Supplementary Note 3: Details of sulfur oxidation.)

It must be noted that because the sulfur signal from pheomelanin is small compared with the signal from disulfide, keratin breakdown reactions may be expected to overprint much of the original sulfur chemistry and complicate spectral deconvolution due to significant peak overlap and background enhancement. Therefore, even though we can resolve heterocyclic sulfur in the fossils, the signal is variable and we conclude that sulfur spectroscopy alone is unlikely to unambiguously resolve pheomelanin residue in ancient speci-mens. However, other elements besides S are associated with pheomelanin.

**Zinc X-ray spectroscopy**. Previous work has shown that Zn is strongly associated with benzosulfur moieties in extant pheome-lanized tissue[1]. Therefore, despite the breakdown of the keratin matrix, if Zn has retained its original coordination chemistry with the residual heterocyclic sulfur compounds within the fossil tis-sues, then pheomelanin pigment chemistry may be resolved through a combination of S and Zn spectroscopy. We performed Zn XAS analyses of fossil material to investigate this important possibility. Figure 4a presents the Zn K-edge XANES analyses for the fossil mouse, an extant *A. sylvaticus*, and standards including red and blonde human hair, Zn bonded to eumelanin, Zn–acetate, Zn sulfate hexahydrate, and inorganic wurtzite (ZnS). Spectra from the fossil fur resemble the extant mouse fur and the human hair. Note that Zn in eumelanin is comparable with the Zn–acetate spectrum. Derivative spectra (Fig. 4b) of inorganic ZnS, red hair, blonde hair, and Zn–eumelanin show that for ZnS, with Zn only coordinated to sulfur, the derivative peak is at a minimum energy. For Zn–eumelanin which has no sulfur, Zn is only light-element coordinated (oxygen or nitrogen) and the peak is at maximum energy. For the red and blonde hair, the derivative peaks are intermediate, indicating a mixed first coordination shell for Zn, including both O and S. The position of the derivative peak for the red hair indicates a higher concentration of sulfur-coordinated Zn than the blonde hair. Analysis of the extended X-ray absorption fine structure (EXAFS) for the Zn compounds (Fig. 4c) indeed corroborates the XANES implications and shows without question that the differences in the absorption spectra are due to the presence of a sulfur backscattering atom in the first

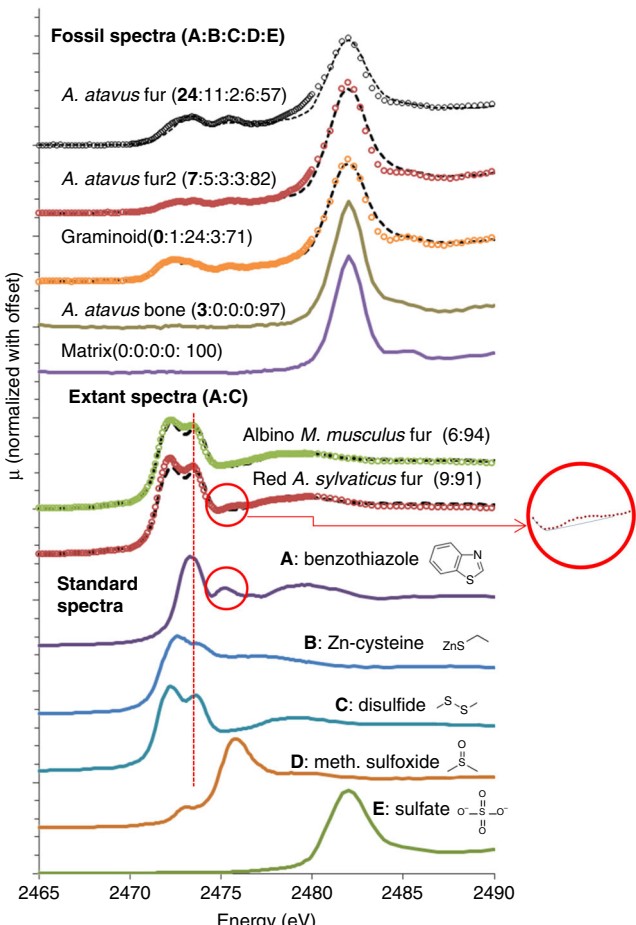

**Fig. 3** Sulfur K-edge XANES data for standard, extant, and fossil specimens. The sulfur standards, **A** benzothiazole, a benzosulfur compound and a key component of pheomelanin, **B** Zn–cysteine, a terminal S organic functional group where we have substituted Zn for H as the exchangeable cation to simulate the configuration expected in pheomelanin tissue or its residue, **C** oxidized glutathione (disulfide) comparator for the dominant sulfur component in keratin, **D** methionine sulfoxide, an oxidized form of organic sulfur, and **E** Zn sulfate. The two extant mice spectra are presented along with linear combination fits (dashed lines) computed as a binary benzothiazole/disulfide system. The red circles highlight the resonance which is strong in the benzothiazole standard and which is resolvable in the red fur. The dashed vertical line furthermore indicates energy where the dominant benzothiazole peak is coincident with the second oscillation in the bifurcated disulfide peak to subtly shift the intensity of that second peak relative to the first—a shift that is discernible in the red fur but not the albino fur. Normalized spectra from the fossil are presented along with LCF fits calculated using all five standards as possible components (ratios of each standard contribution to the fits are given as a set of five numbers in parentheses). Because the fossil bone and sedimentary matrix are almost pure sulfate, for clarity fits have been omitted. Fits are shown as dashed lines for the three soft tissue analyses

shell (see Table 1 for coordination chemistry details. Supplementary Fig. 5 presents all the details of the EXAFS fitting for each spectrum). The summed coordination numbers for the first shell in the two fossil spectra are 5.6 and 5.3, implying that Zn is present in both sixfold and fourfold coordination. The O1 and O2 bond distances are consistent with these two coordination states, where the shorter oxygen bond lengths (i.e., 1.983 and 1.993 Å) imply fourfold oxygen coordination (ideal distance of 1.96 Å) and the longer bonds (i.e., 2.104 and 2.056 Å) are comparable with the range measured for Zn in sixfold oxygen coordination

(2.06–2.184 Å). The sulfur in the first shell of both the fossil and extant fur/hair has bond distances (2.266–2.352 Å) that match well with those determined for Zn coordinated to organosulfur compounds in other studies (2.22–2.34 Å; ref. [31]). Furthermore, distances to the second shell of C atoms in the fossil spectra are also similar to the second shell C distance in blonde human hair, with both fossil and extant spectra indicating a partial coordination configuration comparable with a Zn–carboxylate species as in Zn–acetate. The first shell sulfur coordination number in all of the hair and fur samples is less than one, indicating that all have mixed species of Zn. This detailed XAS analysis shows that Zn–organosulfur compounds are unquestionably present within the fossil fur regions, and furthermore shows that these complexes have a coordination chemistry for Zn that is directly comparable with the Zn coordination environment in pheomelanin-pigmented hair and fur. Figure 4d shows the two model coordination environments for Zn present in both the extant and fossilized soft tissue, with Zn in octahedral coordination with oxygen (or nitrogen) as in eumelanin and in tetrahedral coordination with three oxygens and a single sulfur due to pheomelanin. These results indicate that the correlation of Zn and organic sulfur groups in the fossil mouse maps is not coincidental, but rather the correlation is due to the fact that Zn–organosulfur compounds derived from the original pheomelanin have maintained, at least in part, their original distribution and structure. Therefore, by first confirming via spectroscopy that the Zn has a biochemical affinity, the combined maps of Zn plus organosulfur may then be used to reconstruct pheomelanin pigment distribution in these fossils, as previously shown for extant feathers[1]. The mixed coordination environment for Zn most likely indicates the additional presence of residual eumelanin moieties, which is corroborated by the presence of eumelanin-associated Cu and the correlation of Cu with Zn discussed below. However, the clearly resolved spectroscopic details of Zn–organosulfur complexes (which do not occur in eumelanin) when combined with the mapped distributions of these compounds (which correlate with the residual integument of the organism), provide convincing evidence that the Zn–organosulfur residue is derived from a pheomelanin precursor. These results furthermore support the conclusion that previously resolved organosulfur–Zn complexes in a fossil amphibian larva from the much older Enspel Formation (Oligocene) are also the residue of pheomelanin pigmentation[7]. Barden et al.[7] identified microbodies consistent with fossil pheomelanosomes, showed that the fossil tissue was distinct from the organics disseminated in the sediment, and identified organic-metal chelates within the tadpole residue which can now be seen to be directly comparable with those found here in the Willershausen specimens and in pheomelanin standards[1]. Interestingly, another study[23] of this Oligocene tadpole specimen found eumelanin markers by ToF-SIMS, but could not resolve pheomelanin markers.

**Trace metals, melanin assay, and other analyses**. In order to further test the validity of our results, we performed additional analyses on these specimens and on a range of comparative standards.

One way to test for endogeneity is to quantify the concentrations of elements within fossil material and compare those values with tissues of comparable extant species. If concentrations are anomalously high in the fossil, then the introduction of exogenous material is highly likely. On the other hand, if concentrations are comparable, then it becomes less likely that geochemical processes have reset the chemistry of the specimen. Therefore, to support the XAS results, we completed quantitative

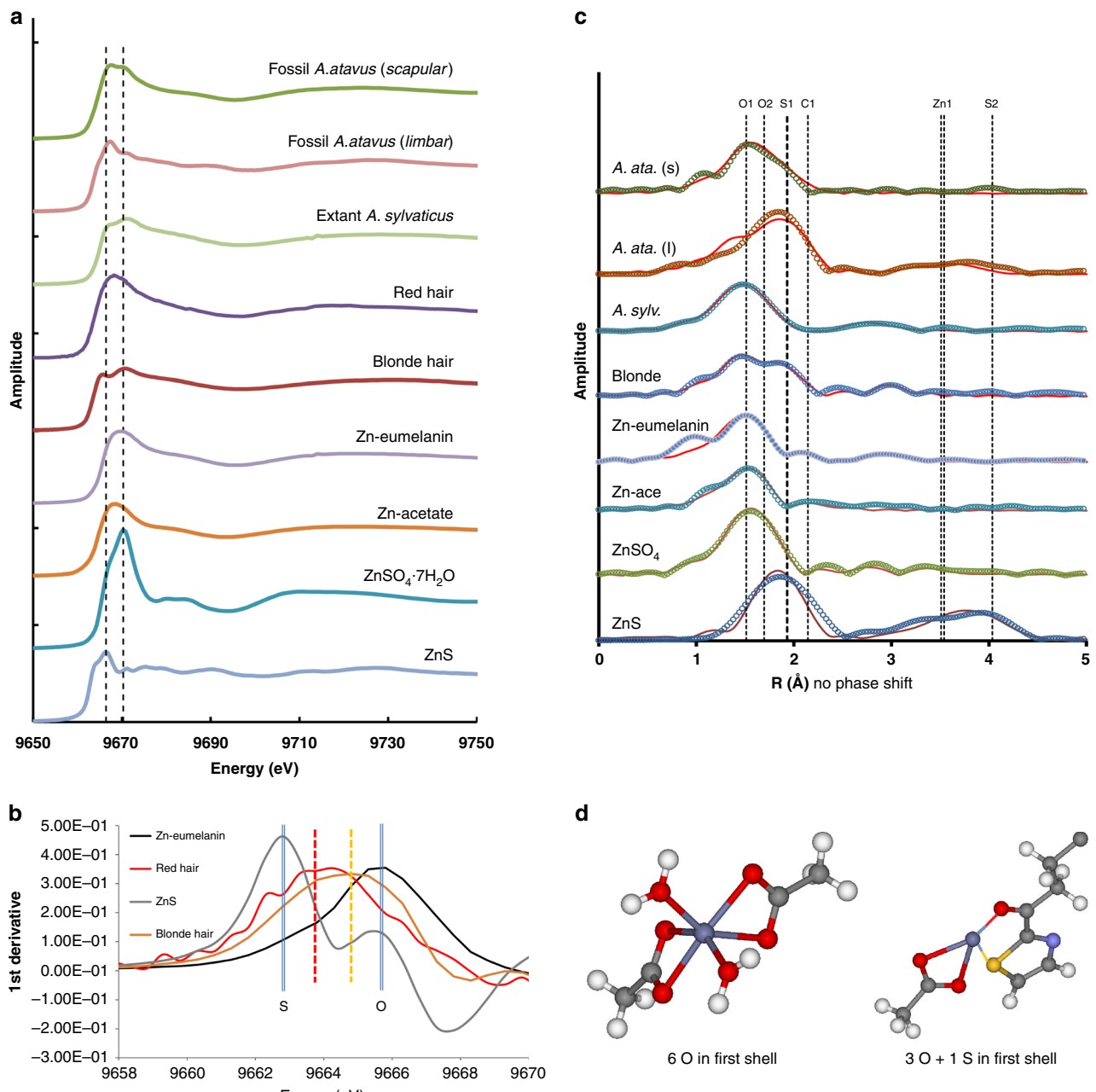

**Fig. 4** Zinc X-ray absorption spectroscopy for fossils and standards. **a** Zinc K-edge XANES spectra for the fossil and extant mouse specimens, human hair, Zn-bonded eumelanin, Zn-acetate heptahydrate, and ZnS. The two vertical lines indicate the absorption spectrum maxima for pure first shell Zn–S and pure first shell Zn–O species. **b** First-derivative analysis in the vicinity of the sulfur "white line" region. The ZnS spectrum is displaced to the low-energy side and the Zn–O spectrum from Zn-substituted eumelanin is displaced toward high energy, with the two pheomelanin bearing hair standards intermediate between the two. **c** Fourier-transformed EXAFS showing relative positions of backscatterers around the Zn central absorber. The heavy dashed line indicates the Zn–S shell. All labels on the vertical lines refer to element shells given in Table 1. Note that the distance values in this figure are not phase shifted and hence are smaller than those presented in the table, but relative positions remain the same. **d** The two Zn coordination environments resolved in both the extant hair and fur and in the fossil mouse fur. The high abundance of organosulfur-bonded Zn in the fur and skin regions of the *A. atavus* fossil is most likely due to an originally high concentration of pheomelanin which is enriched in the tetrahedrally coordinated Zn complex shown in the right of the panel. (Zn = dark gray; C = light gray; O = red; S = yellow; N = blue; H = white)

XRF elemental analyses of extant hair and fur specimens to compare with the fossils. We also completed state-of-the-art melanin assays of the extant material so that we would have an unambiguous dataset of pigment contents for the extant material to assist in understanding and interpreting the relationship between the XAS and XRF data obtained from extant and fossil tissue. Table 2 presents the results of quantitative XRF point

analyses on extant and fossil tissue along with a total melanin and pheomelanin percent assay via AHPO-HPLC for extant hair and fur standards (also see Supplementary Fig. 6; and ref. [32]). Trace metal loading within keratinous integument is complex. Extant specimens for comparison with the fossil data were chosen as representatives of different pigmentation types and are meant to show relative concentrations. The well-known variability in metal

**Table 1 Zn K-edge EXAFS for fossil fur and standard compounds**

| | | A. atavus Fossil DLS-22417 Lumbar | A. atavus Fossil DLS-3705 Scapular | A. syl. Extant DLS-35093 Dark fur | H. sapiens Extant DLS-3631 Blonde | Zn eumel. Standard[1] | Zn acetate Standard DLS-6051 | Ref. Values[48] | ZnSO$_4$.7H$_2$O Standard DLS-6059 | Ref. Values[49] | ZnS Wurtzite DLS-6058 | Ref. Values[50] |
|---|---|---|---|---|---|---|---|---|---|---|---|---|
| O1 | N | 2.9 (4) | 0.9 (2) | 4.2 (6) | 3.6 (3) | 3.9 (6) | 3.7 (1.3) | 2 | 6.0 (4) | 6 | | |
| | R | 1.993 (8) | 1.983 (15) | 1.967 (2) | 1.982 (5) | 1.98 (3) | 1.989 (7) | 1.987 | 2.084 (3) | 2.06–2.12 | | |
| | $\sigma^2$ | 0.011 (2) | 0.010 (2) | 0.011 (1) | 0.011 (1) | 0.012 (4) | 0.012 (2) | | 0.009 (1) | | | |
| O2 | N | 1.7 (2) | 2.8 (3) | | 0.5 (2) | | 0.3 (1) | 4 | 5.6 (1.6) | 4 | | |
| | R | 2.104 (5) | 2.056 (6) | | 2.168 (17) | | 2.034 (25) | 2.184 | 4.16 (0.04) | 4.178 | | |
| | $\sigma^2$ | 0.002 (1) | 0.010 (2) | | 0.002 (2) | | 0.002 (1) | | 0.017 (7) | | | |
| S1 | N | 1.0 (2) | 1.6 (4) | 0.3 (1) | 0.6 (2) | | | | | | 4.3 (4) | 4 |
| | R | 2.352 (5) | 2.315 (11) | 2.266 (14) | 2.306 (6) | | | | | | 2.330 (4) | 2.341 |
| | $\sigma^2$ | 0.001 (1) | 0.015 (3) | 0.006 (2) | 0.001 (1) | | | | | | 0.008 (1) | |
| C1 | N | 4.9 (6) | | | 1.8 (8) | 1.7 (8) | 0.9 (5) | 2 | | | | |
| | R | 2.66 (1) | | | 2.66 (5) | 2.72 (5) | 2.688 (21) | 2.552 | | | | |
| | $\sigma^2$ | 0.011 (3) | | | 0.022 (18) | 0.016 (1) | 0.001$^a$ | | | | | |
| O3 | N | 11 (6) | | 1.5 (2) | 1.6 (7) | | | | 0.7 (3) | 24 | | |
| | R | ms 4.11 (5) | | 3.25 (1) | 3.313 (15) | | | | ms 3.25 (4) | 3.5–3.7 | | |
| | $\sigma^2$ | 0.045 (24) | | 0.001$^a$ | 0.004 (3) | | | | 0.001$^a$ | | | |
| Zn1 | N | 9 (3) | | | | 1.5 (8) | | | | | 12 (3) | 12 |
| | R | 3.85 (2) | | | | 3.26 (6) | | | | | 3.848 (7) | 3.823 |
| | $\sigma^2$ | 0.019 (4) | | | | 0.014 (6) | | | | | 0.018 (3) | |
| S2 | N | | | | | | | | | | 10 (2) | 9 |
| | R | | | | | | | | | | 4.507 (21) | 4.483 |
| | $\sigma^2$ | | | | | | | | | | 0.019 (3) | |
| S% in the first shell | | 18 | 30 | 7 | 13 | 0 | 0 | | 0 | | 100 | |
| $S_0^{2 b}$ | | 0.79 | 0.79 | 0.9 | 0.78 | 1.02 | 0.81 | | 0.98 | | 0.9 | |
| $\chi^2_\nu$ $^c$ | | 5.5 | 4.51 | 0.58 | 7.5 | 47.34 | 17.4 | | 17.29 | | 2.12 | |
| $\Delta E_0$ $^d$ | | 3.46 | 2.46 | 1.07 | 4.64 | −0.14 | 0.4 | | 1.09 | | 3.23 | |
| $R$ $^e$ | | 0.023 | 0.031 | 0.006 | 0.039 | 0.022 | 0.053 | | 0.02 | | 0.017 | |

$^a$A parameter held fixed during fitting, and ms denotes multiple scattering. 1$\sigma$ errors given in parentheses (as ± values on terminal digit[s]). See ref. [1] for more pheomelanin EXAFS
$^b S_0^2$ is the amplitude reduction factor
$^c \chi^2_\nu$ is reduced chi-square
$^d \Delta E_0$ is edge position shift relative to theoretical value
$^e R$ is a goodness-of-fit parameter

**Table 2 XRF point analyses and melanin concentrations (units ppmW except where otherwise indicated$^a$)**

| | H. sapiens Brown | Blonde | Red | A. sylvaticus Dark red | White | M. musculus Dark gray | Albino | A. atavus Dorsal fossil | A. atavus Lateral fossil |
|---|---|---|---|---|---|---|---|---|---|
| Ca | 843 (34) | 994 | 387 | 2163 | 1530 | 961 | 297 | $^b$ | $^b$ |
| Ti | 4 (1) | 25 | bld | 14 | 7 | | 4 | 1883 | 1768 |
| V | 1.0 (4) | 0.5 | 2 | 3 | 1 | | 2 | 400.5 | 564 |
| Mn | 2.0 (8) | 2 | 0 | 3 | 1 | | 1 | $^b$ | $^b$ |
| Fe | 36 (5) | 145 | 10 | 417 | 207 | | 61 | $^b$ | $^b$ |
| Ni | 10 (3) | 4 | 1 | bld | bld | | bld | 309 | 57 |
| Cu | 251 (8) | 93 | 10 | 3 | 7 | 24 | 10 | 74 | 21 |
| Zn | 1250 (50) | 670 | 102 | 74 | 69 | 154 | 105 | 1530 | 2480 |
| As | 0 | 0 | 5 | 1 | bld | | bld | 72 | 64 |
| Zn/Cu | 4.9 | 7.0 | 10.2 | 24.7 | 9.9 | 6.4 | 10.5 | 20.7 | 38.8 |
| TM (µg/mg) | 14.5 (2) | 0.89 (9) | 7.2 (6) | 84.5 (8) | | | | | |
| EM (µg/mg) | 12.9 (2) | 0.29 (6) | 3.4 (9) | 52.5 (4) | | | | | |
| PM(BT) (µg/mg) | 0.02 (1) | 0.050 (3) | 0.5 (3) | 1.72 (3) | | | | | |
| PM(BZ) (µg/mg) | 1.6 (1) | 0.55 (5) | 3.4 (1.2) | 30.3 (4) | | | | | |
| PM(BT)/TM | 0.14% | 6.12% | 6.27% | 2.03% | | | | | |
| PM(BZ)/TM | 11.03% | 61.80% | 43.70% | **35.86%** | | | | | |

$^a$2$\sigma$ errors on concentrations due to counting statistics range from 4% (e.g., Ca, Zn) to 40% (e.g., Mn) relative, with indicative errors for the concentration ranges presented here given in the first column as errors on the terminal digits in parentheses
$^b$Not discriminated from sedimentary matrix background. Point analyses from SSRL beamline 6-2 and DLS beamline I18. Standard error for the terminal digits of the melanin data also shown in parentheses
TM = EM + PM(BT) + PM(BZ). TM indicates total melanin, EM is eumelanin
PM(BT): Pheomelanin derived from benzothiazine (BT) units, as determined by hydroiodic acid hydrolysis – HPLC
PM(BZ): Pheomelanin derived from benzothiazole (BZ) units, as determined by AHPO – HPLC [24]

concentrations in feathers, hair, fingernails, and other tissues is precisely why elemental mapping is so crucial and why bulk metal analysis can be misleading with respect to integument. For these reasons, we also present ratios of Zn/Cu in Table 2 in an attempt to account for the range in concentrations observed in extant and fossil tissue.

Ca, Fe, Cu, and Zn are the most important metals chelated by melanin, with the first three strongly chelated by eumelanin[1]. As shown in Table 2, the brown human hair is rich in eumelanin (88.9%) and is relatively high in three of these four trace metals because eumelanin is an efficient chelator. Total concentrations are much lower in the red human hair; however, the extant *A. sylvaticus* fur shows relatively high trace metal concentrations in both dark- and light-pigmented regions, suggesting that even the light areas contain traces of pigment. Both extant murid specimens have higher transition element concentrations in darker pigmented fur. Cu, when associated with organic compounds, has been shown to be an indicator of eumelanin distribution[2]. However, it is clear that the main trace metal difference in the extant *A. sylvaticus* between light and dark regions is the Zn/Cu ratio, which is much higher in the pigmented, pheomelanin-rich region. This is consistent with previous data on extant integument[1] and with the Zn XAS and XRF data for the extant material presented here, both of which resolve an affinity between Zn and pheomelanin-associated organic sulfur. For example, Supplementary Fig. 7 shows analyses of red human hair with Zn dominating the metal loading. (Supplementary Fig. 8 presents details of additional XRF point analyses.) If we then consider the concentration results from the extant tissues, the high Zn/Cu ratios in the integuments of both fossil *A. atavus* specimens (Table 2), the similarities in the Zn–EXAFS of the fossils to known pheomelanin-rich tissue (Table 1), plus the relatively good correlations between organic S and Zn (Supplementary Fig. 2), all evidence points toward the organosulfur–Zn compounds within the fossil being derived from a pheomelanin precursor. Furthermore, the correlation between Cu and Zn presented in Supplementary Fig. 2c, d when combined with the mixed first shell coordination environment for Zn suggests that the fossil residue is derived from a mixture of both original eumelanin and pheomelanin, implying that *A. atavus* would be best described as a mixed melanic phenotype. (For comparison to the fossil maps, SRS-XRF maps of a dark gray *M. musculus* are shown in Supplementary Fig. 9.)

Although point analyses show that concentrations are elevated for several trace metals in the fossil integument, concentrations are not as high as would be expected for mineral precipitates (several weight percent). In fact, considering that the fossil tissue has desiccated, the single-element concentration maximum of 2480 ppm in the fossil compares well with the extant maximum of 2163 ppm. It is thus clear from the XRF point analysis that the Zn, Cu, and S concentrations in the fossil do not require postmortem enrichment. Indeed, XRF analyses of light elements (Supplementary Table 1) show that the sulfur concentration in the fossil is nearly identical to sulfur concentrations in extant fur, and therefore complete geochemical replacement of the original fossil is extremely unlikely; much more likely is hydrolysis and oxidation of the original biomaterial in situ, as the sulfur spectroscopy indicates.

Direct melanin analysis shows indeed that the extant *A. sylvaticus* is a eumelanic mixed phenotype, however with significant quantities of pheomelanin [PM(BT) + PM(BZ) = 32.02 μg/mg]. Pheomelanin derived from benzothiazole units dominates the pheomelanin inventory in all of the extant specimens, consistent with the S XANES analyses. Furthermore, the high absolute quantities of pheomelanin within *A. sylvaticus* indicate that there are abundant pigment-associated S-binding

sites for Zn in the extant material. Thus, even if a large fraction of these benzothiazole units were destroyed by aging, there would still be sufficient S-binding sites to produce the mixed Zn coordination environment observed in the *A. atavus* fossil specimens. Therefore, the trace metal point analyses and melanin pigment assay are consistent with the mapping and spectroscopy, indicating a relationship between sulfur-bonded Zn and pheomelanin in the extant tissue, as well as confirming the presence of a residue in the fossil that is most probably derived from both types of melanin pigments. The melanin assay also explains why the Zn K-edge derivative peak for red hair is shifted so far toward the ZnS standard in Fig. 4b: there is over six times as much heterocyclic sulfur in the red hair than is present in the blonde hair.

Fourier transform-infrared (FTIR) spectroscopy (Supplementary Fig. 10) was also completed in order to further test the endogeneity of the chemistry of the fossil soft tissue. The fossil soft tissue FTIR spectra are completely different from the background matrix (see Supplementary Fig. 10), containing clear strong organic absorption bands including amide I–III, hydrocarbon, and C = O. The matrix is however dominated by carbonate and silicate absorption bands and includes only a single weak absorption band for trace organics (C = O). In the fossil, the amide bands would be expected as a breakdown product of alpha keratin from both skin and fur, and the hydrocarbons originate from the lipid and other aliphatic components in the original organism. The C = O moiety is consistent with the presence of fatty acids derived from the original biochemistry and may provide some of the oxygen-terminated functional groups which we see via EXAFS bonded to the trace metals. We also note that the original pheomelanin is rich in carboxylic functional groups due to the presence of abundant arginine and therefore at least part of the C = O groups may also be derived from the pheomelanin pigment itself. These results clearly show the presence of amide groups within the fossil, thus resolving organic residue that is indicative of integument material and is consistent with the X-ray imaging and spectroscopy, as also shown in several previous studies of extant and exceptionally preserved fossil integument (e.g., refs. [2,13,14,33]).

Environmental scanning electron microscopy (ESEM) was applied in order to determine whether melanosomal microstructures might be present within the integument residue. Other studies have based the reconstruction of pigment distribution on putative fossilized melanosomic structures, with rod-shaped structures approximately 0.5–2 μm in length classified as fossil eumelanosomes and spheroidal structures with a diameter of approximately 1 μm classified as fossil pheomelanosomes[22,34–37]. Doubts about the interpretation of these structures have been raised due to their similarity to microbial fossil products[25]. The structural approach also has limitations due to the fact that it relies on electron microscopy and therefore only examines a tiny fraction of the surface area of fossil material requiring extrapolation to whole-organism color reconstruction. We furthermore note that in all but the most extreme cases, both types of melanosome will be found together: see Table 2 for the proportions of melanin pigments in extant brown, blonde, and red human hair. Even the light blonde hair has over 30% eumelanin pigment, and so simply resolving eumelanosomes or pheomelanosomes in a fossil specimen of hair such as this would not unequivocally reveal specimen pigment predominance.

ESEM revealed the presence of numerous spheroidal bodies (0.8–2 μm; see Supplementary Fig. 11), calcite rhombs, phyllosilicate platelets, abraded sediment grains (quartz and feldspar), and Ti-rich aquatic microbodies (chrysophyte cysts: ~30 μm diameter). ESEM energy-dispersive spectrometry (EDS) revealed

high Ca concentrations in the matrix, consistent with the XRF analyses, the XRD results, and the presence of calcite rhombs. The abundant micron-sized spheroidal bodies blanket much of the fur and skin regions of the fossil *A. atavus* (see Supplementary Table 3 and Supplementary Fig. 11). The corrugated surfaces, shapes, and size range of these are consistent with fossilized melanosomes. Destructive analysis of these precious specimens was not allowed; therefore, we could not scrape away the bodies in order to determine whether they are penetrative or analyze them in isolation from the underlying material. Thus, we simply point out that these spheroidal bodies do indeed correlate with the fossil and are absent from the matrix bedding plane, hence they are definitely associated with the fossil.

Similar microbodies were found to correlate with organosulfur–Zn complexes in a postulated pheomelanin-pigmented amphibian[7]. However, it should be made clear that the Zn spectroscopy and fluorescence imaging presented here would not be compromised if these microbodies were reclassified as microbes via some other analytical approach at a later date, because the Zn–EXAFS is determined by fluorescence with an attenuation depth of approximately 40 μm. Therefore, any surface film of microbodies would contribute far less than 5% of the fluoresced Zn X-rays and most likely would not significantly contribute to the signal from the underlying mouse fossil. In any case, the presence of these micron-sized spheroidal bodies which dominate the exposed fur and skin regions of the fossil *A. atavus* (see Supplementary Fig. 11a–d) suggests that fossil pheomelanosomes may be present, which further supports the chemical evidence for the presence of pheomelanin residue.

Finally, in order to improve the spatial resolution of our X-ray imaging so that we could be certain that the distribution of residual chemistry displayed biological structural control, we completed microfocus XRF mapping (Supplementary Fig. 11e–g). This showed that the Zn distribution resolved hair-like filaments within the *A. atavus* specimens and that lineations on the periphery grade into masses of higher Zn concentration from distal to proximate locations, as would be expected if this metal pattern has been derived from original fur. Our results therefore go beyond the microstructures and allow a broad spectrum of the chemical inventory of the integument to be used to reconstruct the original distribution of organismal biochemistry. We have also mapped the elemental inventories of two other specimens sampled from the Willershausen locality, including an intact single feather and a partial, but articulated, frog. Patterns in both of these fossils related to soft tissue chemistry can be resolved, however, neither of these fossils displays the same type of correlation between zinc and organic sulfur as shown by the *A. atavus* specimens. This supports the hypothesis that the patterning seen in these mouse fossils is endogenous and is not a generic feature of the taphonomy at Willershausen.

For specimens that are ~3 million years old, we cannot a priori dismiss the possibility that geochemical processes may have altered the distribution of Zn and S. However, our results indicate that for these specimens, this is unlikely, given that Zn and organosulfur compounds map discretely within the fur and any postulated geochemical process would have to deposit organo-sulfur compounds of Zn in such a way as to mimic the original organism's structure. All of the sulfur concentration measurements via synchrotron XRF and ESEM-EDS (Supplementary Fig. 8, Supplementary Tables 1 and 3) have an average of 5.2% (range: 1.14–7.75, $n = 9$) comparable with the sulfur content measured for extant mouse fur (5.51%, Supplementary Table 1) and therefore there is no requirement for mass transfer of sulfur into this system to account for the observed sulfur content. Sulfurization has been considered, but given that there is no sulfur enrichment in the tissue analyzed in our study and no such

sulfurization is determined for other Willershausen specimens[38], it is highly improbable that sulfurization is responsible for the distribution of organic sulfur in our two specimens. Likewise, the range of melanin-associated trace metal concentrations within the fossil fur (21–2480 ppm) is comparable with the range exhibited within extant hair and fur (10–2163 ppm) and therefore an influx of metals via geochemical fluids into the system is not required to account for the chemistry we resolve. Small Ti-rich cysts are present, but these are discrete structures which are easily discriminated from the residual fur. Furthermore, the precipitation of inorganic Zn minerals, the most likely process to introduce Zn into the fossilized integument, is wholly inconsistent with the XAS spectroscopy.

## Discussion

Our results show that the Zn coordination chemistry in these exceptionally preserved fossils is closely comparable with Zn within modern pheomelanin-pigmented hair. Both fossil specimens exhibit the same Zn/S coordination chemistry in the residual integument as well as the same Zn–organosulfur compound distribution pattern. Sulfur spectroscopy of fur and hair has revealed a discrete signature caused by heterocyclic organic sulfur compounds that can be used as a biomarker for the presence of pheomelanin pigment in pristine soft tissue. However, due to sulfur oxidation reactions within fossilized integument, this single-element marker becomes progressively obscured and may not be resolved in fossil specimens. Indeed, the high-sulfur content of keratinous tissue (~7%) will mean that as the disulfide bonds break and oxidize, the resulting increased background in the sulfur K-edge spectra will eventually occlude the relatively weak XANES oscillation diagnostic of heterocyclic sulfur. However, it has been shown that in extant tissue, Zn is not complexed with keratin but is almost entirely bound to pheomelanin pigment by a Zn–S bond[1]. Our findings indicate that this Zn–S bond appears to be more stable than the disulfide bond in keratin. This explains why our observed Zn–EXAFS signal contains significant amounts of residual Zn–organosulfur compounds and these are resolvable, despite the degradation of the surrounding keratin. Thus, a combination of Zn spectroscopy with Zn and organic sulfur XRF mapping resolves and maps the distribution of pheomelanin-derived Zn complexes. The mixed coordination of Zn plus the correlation between Zn and Cu in the fossil integument indicates the presence of and similar distributions of both eumelanin and pheomelanin.

Finally, how do these findings impact our understanding of this extinct *Apodemus* species? The uneven distribution of Zn, Cu, and organic S suggests that the integument may not have been uniformly melanized. There is a lack of organically bound Zn in the extremities (i.e., tail and feet) of both *A. atavus* specimens as shown in the XRF images. Optical photographs show clearly that this is not due to preferential separation between part and counterpart, because the optical images show comparable quantities of soft tissue residue on each side of the bedding plane fossils (see Supplementary Fig. 12). The lower quantities of organosulfur–Zn complexes in these regions compared with the corpus could reasonably be caused by lower in vivo quantities of pigment in these parts of the organism. Given that our data are consistent with regions of low-pigment densities in analogous parts of extant-related species (e.g., *A. flavicollis*), we postulate that these areas of *Apodemus atavus* could also have been weakly pigmented. However, we also note that relative to a third specimen of *Apodemus atavus*[27], soft tissue preservation in the tail regions of both of the specimens imaged here is less complete, and therefore differential preservation of soft tissue may be acting to complicate the assignment of original pigment patternation.

The correlation maps and the related spectroscopy from both fossil specimens however do indicate with a high degree of

certainty that the corpus of this species was rich in both melanin pigments as in *A. sylvaticus*. Zoning of organosulfur–Zn complexes within soft tissue residue may now provide a reliable chemical method for resolving pheomelanin pigment residue in extinct organisms. This study demonstrates that the spatial distribution of different forms of melanin residues in extinct organisms may be resolved nondestructively over large areas (dm²) even after 3 million years of degradation.

## Methods

**Fossil specimens**. *Apodemus atavus* specimens GZG.W.20027b (holotype) and GZG.W.17393a from the Willershausen conservation lagerstätte were loaned to PLM for this study by the University of Göttingen.

**Extant mouse specimens**. An intact, whole-organism specimen of an albino *M. musculus* was purchased from an approved commercial supplier. The dark gray *M. musculus* and the red *A. sylvaticus* were wild deceased specimens recovered from pest control. This study did not involve research with live animals. All animal tissues were either fossilized or acquired from already-deceased specimens and provided via ethical suppliers. Before analysis, fur samples were freeze-dried and then stored in aluminum foil packets. No further treatment was applied to the fur.

**Hair samples**. Hair samples of approximately 10 g were donated by individuals with typical natural blonde, red, and brunette pigmentation (see below). Samples were rinsed in deionized water to remove adventitious dust but were otherwise untreated. None of the samples had been exposed to dyes or any other aggressive hair treatment processes.

The rare and precious nature of the fossil specimens meant that we could not apply any destructive sampling techniques and therefore the nondestructive nature of our research design was of critical importance. We did however perform destructive analyses using high-performance liquid chromatography (HPLC) on extant hair and fur specimens for comparison with the fossils. Full details on measurement conditions in this study are available below.

**Synchrotron methods**. Our analytical methodology[1–15] makes extensive use of nondestructive synchrotron-based methods, including SRS-XRF, XAS, microfocus XRF imaging, and quantitative synchrotron XRF point analyses.

SRS-XRF imaging was performed at the Stanford Synchrotron Radiation Lightsource (SSRL) wiggler beamline 6-2 at the Stanford Linear Accelerator Center (SLAC, CA, USA). Extensive and detailed descriptions of SRS-XRF mapping applied to fossils are provided in recent previous publications[2,12,15] and so are only summarized here. Experiments were operated with an incident beam energy of either 13.5 keV (flux calculated between $10^{10}$ and $10^{11}$ photons $s^{-1}$) or 3.15 keV (flux ~$10^9$ photons $s^{-1}$) and a beam diameter of either 50 or 100 μm defined by a pinhole. Fluoresced X-rays were detected using a single element Vortex silicon drift detector. SRS-XRF maps from SSRL were processed from the raw detector count raster files using a custom MATLAB computer script that converted the data array into viewable 8-bit tiff images clipped at various contrast percentiles. Image subtraction was performed in ImageJ via the built-in image subtraction function and the image correlation was completed using the CorrelationJ[39] plugin.

Microfocus imaging was completed at beamline I18 at the Diamond Light Source (DLS), which allows for small-scale elemental mapping (mm²) at micron resolution. The combination of Kirkpatrick–Baez mirrors, a double-crystal Si(111) monochromator, and a 4-element Vortex silicon drift detector allow for a full EDS spectrum to be recorded for each pixel of the elemental map and for the collection of full EXAFS. Flux was estimated to range between $10^{11}$ and $10^{12}$ photons $s^{-1}$. Maps were processed using the ROI imaging tool in PyMCA freeware[40] by defining the X-ray emission energy of an element in the recorded EDS spectra.

Energy-dispersive spectra for quantification were obtained from single points at SSRL and DLS. SSRL spectra were collected at a single point (50 μm diameter) for 100 live seconds. DLS point spectra were also collected at a single point (5 μm diameter) for 30 live seconds. All EDS spectra were fit using PyMCA[40] from fundamental parameters of the experiment using a Durango apatite (fluoroapatite) mineral standard with known element concentrations for calibration. 2σ errors on concentration were calculated using the standard deviation on peak area for each element output by PyMCA (e.g., Supplementary Fig. 4). Please consult[10] for further details. The concentrations reported are an average of three individual EDS spectra taken within a few hundred microns of each other on the same specimen.

Sulfur XANES data were recorded at SSRL beamline 6-2 in fluorescence mode using a single element Vortex silicon drift detector set at ~60° scattering angle. S XANES were also collected at DLS microfocus beamline I18 in fluorescence mode using a four-element Si drift Vortex fluorescence detector set at 90° scattering angle using a defocussed beam with slit apertures of 500 μm (vertical) and 50 μm (horizontal). In both cases, the samples were placed in He-purged chambers to minimize air absorption of the fluoresced signal. Beam-induced photooxidation was monitored and minimized by using a defocused beam, rapid scan speeds, and

by slightly moving the beam between scans to refresh the area analyzed. Zn EXAFS was collected at DLS beamline I18 in fluorescence mode using a four-element Si drift Vortex fluorescence detector set at 90° scattering angle, slit aperture same as above. Zn and S standards were used to calibrate the energy of the monochromator position. Data averaging, background subtraction, data normalization, and fitting of the EXAFS spectra were performed using the SIXpack software package[41]. In the EXAFS fits, a statistically significant improvement in a fit was defined by a $\chi^2_\nu$ critical value, where

$$\chi^2_\nu(\mathrm{crit}) = 1 + 2\sqrt{\frac{2}{\nu}}. \tag{1}$$

If the addition of the path decreased the reduced $\chi^2$ by more than this critical value, then the additional path was judged to be significant[42]. Fitting was done in R-space over variable ranges of parameter space depending on data quality: the regions fit are presented for all materials in Supplementary Fig. 5 below, except for the Zn–eumelanin which is given in ref. [1]. After a reasonable fit was calculated and the maximum number of statistically significant paths was determined, confidence limits were calculated for each fit parameter consecutively by allowing each variable to float as a guess within FEFFIT as accessed via the Advanced EXAFS Fitting portion of the SixPack program[43]. XANES spectral analysis and LCF was accomplished using the Athena programme, which is part of the Demeter suite of XAS analytical tools[44]. Spectral energies were calibrated with standards and spectra were normalized prior to LCF calculations. For XAS analyses, S standard compounds were mixed with boron nitride to obtain a S concentration of ~5 wt% to minimize self-absorption effects. Cu XANES spectra were also obtained at I18 for the dorsal fossil and for the brunette hair; however, beam-induced photoreduction compromised these measurements. Supplementary Table 2 provides details of sulfur standards.

**Environmental scanning electron microscopy**. ESEM was completed using a Zeiss Supra40 instrument at the DRIAM Analytical Service, Dalton Research Institute, Manchester Metropolitan University. Accelerating voltage of 1 kV was used at working distances of 3–5 mm. Secondary electron images were collected, and for the images presented here, absolutely no gold or carbon coating was used. All samples were analyzed without any chemical or physical alteration to their surface. Standard clean laboratory sampling handling protocols were used throughout. Point analyses of elements present were acquired using standardless energy-dispersive spectroscopy (Oxford Instruments) at 20 kV and a working distance of 15 mm. Spectra collected were fitted and elements quantified using proprietary ZAF-corrected software. Counting times in excess of 100 live seconds were performed to maximize the signal-to-noise ratio of the spectra.

**X-ray diffraction**. Grazing-incidence XRD scans of the lateral specimen surface were run on a Bruker D8 Advance diffractometer using a fixed incident angle of 2.5°, with a detector step size of 0.02° at a speed of 2 s step⁻¹. A Goebel mirror attachment provided parallel X-ray beam optics, and a scintillation detector with a Soller slit assembly was used. Incident beam wavelength was 1.5418 Å (Cu Kα). Peak assignment was achieved using Eva 14.0 software to compare the measured data with standards from the ICDD Powder Diffraction File.

**Imaging Fourier transform-infrared spectroscopy**. Infrared spectra were collected using a Perkin-Elmer Spotlight 400 instrument with 4 cm⁻¹ resolution. All spectra were background subtracted and specimens required no further preparation.

**High-performance liquid chromatography**. We used the best available destructive methodology for quantifying the eumelanin and pheomelanin concentrations in the human hair standards used for EXAFS comparison with the fossil and extant specimens[32,45,46]. Melanin identification and quantification was performed as follows. Totally, 2–18 mg of hair or fur were homogenized with Ten-Broeck homogenizer at a concentration of 10 mg/mL $H_2O$ and 100-μL (1-mg) aliquots were subjected to AHPO[26] and hydroiodic acid hydrolysis[47]. Brown hair and *A. sylvaticus* data are averaged from duplicates measured on one sample, red and blonde hair data are averaged from duplicates measured from two subsamples taken from the same larger sample (average of four measurements).

**Reporting summary**. Further information on research design is available in the Nature Research Reporting Summary linked to this article.

## Data availability

The data that support the findings of this study are available from the corresponding author upon reasonable request. The source data files for all figures and tables are listed in a source data file.

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

## Acknowledgements

The University of Göttingen graciously loaned us the fossil specimens. Funding was provided by a UK Natural Environment Research Council grant NE/J023426/1. Portions of this research were carried out at the Stanford Synchrotron Radiation Lightsource (CA, USA), a national user facility operated by Stanford University on behalf of the U.S. Department of Energy, Office of Basic Energy Sciences. Portions of this research were also carried out at Diamond Light Source, beamline I18 (UK, allocations SP12948, SP11865, SP9488, SP8597, and SP7749). We thank support staff at SSRL and DLS. PLM thanks the Science and Technology Facilities Council for their support (ST/M001814/1). We are grateful for access to the DRIAM Analytical Service, Dalton Research Institute, Manchester Metropolitan University.

## Author contributions

N.P.E., A.vV., J.A., P.L.M., U.B., W.I.S., V.M.E., D.S., R.A.-M., K.I., B.E.vD., F.K., and R.A.W. all participated in the synchrotron analyses. N.P.E., P.L.M., and R.A.W. composed the experimental design. K.W. and S.I. conducted the melanin identification and quantification experiments. R.A.W. analyzed and fit all of the synchrotron data. P.L.M., N.P.E., and R.A.W. composed the paper. All co-authors gave critical input to the text.

## Additional information

**Competing interests:** The authors declare no competing interests.

