## [Peer Review File · Nature Communications]

Reviewers' Comments:

Reviewer #1:

Remarks to the Author:

In the manuscript entitled "Pheomelanin pigment remnants mapped in fossils of extinct mammals" the authors (P.L. Manning, N.P. Edwards, U. Bergmann, J. Anné, W.I. Sellers, A. van Veelen, D. Sokaras, V. M. Egerton, R. Alonso-Mori, K. Ignatyev, B. van Dongen, K. Wakamatsu, S. Ito, F. Knoll and R.A. Wogelius) report on the development/validation of a chemical imaging methodology to map melanin pigments, and in the specific pheomelanins, in extant and fossil soft tissues.

This work is part of an up-to-date and very challenging research carried out by the authors and focused on the identification/localization of melanin pigments in extinct organisms, with the aim of adding important tiles for delineating the evolution pathways of some species.

The solid consciousness of the state of the art, the multidisciplinary expertise, the careful and meticulous approach and the innovative and sophisticated techniques employed to carry out this work, represent the real success of this paper.

The work is original and the topic is of interest for the audience of Nature Communications, the bibliography is accurate, the data are well presented and the conclusions are adequately discussed.

For these reasons I strongly recommend this paper for the publication on Nature Communications as is.

Reviewer #2:

Remarks to the Author:

The manuscript "Pheomelanin pigment remnants mapped..." submitted to Nature Communications is a comprehensive study on the trace metal distribution of a fossil mammal, suggesting a relationship with residual melanin pigments.

The result that the 3 million year old field mouse *Apodemus* was brown colored by pheomelanin and eumelanin similar than present-day *Apodemus* is not very surprising. However, the present study is interesting, because of the possible implications for the reconstruction of pigment patterns in extinct organisms in general.

The analytical work is very detailed and consistent. Although Zn elemental mapping of fossil specimens by SRS-XRF and XANES analysis as an approach to reveal the spatial distribution of melanins has already reported by ref. 7 (Barden et al., 2015), this study includes also the element sulfur. Limitations of the methodology would be that Zn occurs both in pheomelanin and eumelanin (although with a higher Zn/Cu ratio for pheomelanin) and other sulfur organic compounds may be present in fossils.

My major concern about this manuscript regards the reconstructed pigment distribution of *Apodemus atavus*.

It should be considered that the depicted fossil specimens don't show the complete fossils. Generally, one part of fossils from Willershausen is preserved on a slab, while another part of the fossil remains on the counter slab. However, no information regarding the counter slabs of the two specimens is given in the manuscript. Both counter slab fossils (GZG.W.20027A and GZG.W.52-17393a) have been reported in the original description of the *Apodemus* specimens from Willershausen (S. Rietschel, G. Storch. *Senckenbergiana Lethaea* 54, 491-519 (1974)). This paper should be cited in the main article,

not in the supplementary part. Moreover, some material may have been removed during former preparation of the fossils. It is obvious that some parts of the tissue surface are lost or not considered in the elementary mapping (especially in the specimen depicted in Fig. 1), because otherwise also the bones of the fossil would be covered with tissue material and fur. In the specimen depicted in Fig. 1 no tissue material and fur is preserved in the tail (only bones!) and most parts of the feet and the jaw, thus no statement regarding the pigmentation of these parts is possible (last paragraph of page 9, and life reconstruction). Because of this partial loss of material on the analyzed slabs, the layer and thickness of the preserved tissue material and fur is not the same in different parts of the fossil. In my view, this explains to a significant extent the uneven distribution of Zn, Cu, and S, especially of the specimen depicted in Fig. 1. Although theoretically resolving of pigment patterns may be possible for other fossils, contrary to the authors, I see no evidence for pigment patterning of *Apodemus atavus*.

Reviewer #3:

Remarks to the Author:

The major claim of the paper is that the authors have identified pigments in fossil mammals. They are careful in the title of the paper to avoid that claim that they have identified the actual pigments, rather they have identified "remnants" of the pigments. However, the language of the paper very quickly reports the finding of pigments often without caveats.

For me this is where the major challenge of this field begins and ends. The organic pigments they refer to are pheomelanins, which as I understand are related to melanins, which are notoriously difficult to define structurally even in modern tissues. Thus, it is not the organic constituents that are identified but inorganic elements that are claimed to be specifically associated with these pigments. The claims of the paper rests on the purported specificity of these associations.

The introduction provides a long summary of the field and critiques other studies drawing attention to the limitations of the TOF-SIMS method, then proposes a new approach based on chemical imaging methodology. They say this "show[s] that organosulfur-Zn complexes are indicators of pheomelanin (red pigment)". I looked back at several of the cited papers which confirmed that at no point are the organic pigments ever structurally unambiguously identified. The presence of pigments are always inferred from the distributions of various inorganic elements, which the authors have found to be associated with pigments in modern fur and feathers. Given the latter they authors then use the occurrence of these inorganic elements to map the occurrence of different coloured pigments on fossils - this is quite a leap of faith given the complexities of fossilisation systems. One paper I looked at used various inorganic elements to map the pigments a 150 million year old bird feather. The curious thing about the images they showed is that the visible light image actually appeared to show a clearer representation of the feather than the element maps. I guess the authors would argue that the blurring in the element images is caused by some sort of diffusion during fossilisation - this made me feel uncomfortable - perhaps the elements arose diagenetically by diffusion.

The paper is very long and extremely complex in its explanation of the various spectroscopic techniques used wrapped up with the detailed morphological descriptions. For this paper to be publishable I feel it needs to be shortened and the language simplified. I was left asking why the explanations were so complex. I am suspecting as this is a controversial area which is clearly energising a number of groups and the rivalries are quite strong and so the high level of detail is included to head off criticism.

The basic problem as I see it is that the various workers in this field are wrestling with a seemingly insurmountable conundrum - the pigments are difficult to define in modern animals and identification

in fossils made all the worse by fossilisation, thus rather than identify the pigments they use non-specific tracers which might in themselves be impacted upon by fossilisation. I'm not sure I see how they can break out of this.

Arguments related to the so-called fossil organic S also become extremely circular. Sulfurization of organic matter is a well know phenomenon in diagenesis and thus the extensive presence of S is to be expected. And if the right elements are present then S metal complexes would be entirely expected but how can these be distinguished from endogenous S metal complexes - this would seem to be another conundrum?

Overall, the paper contains some wonderful images, great spectroscopy but there is considerable circularity in the arguments made for the use of non specific inorganic elements as proxies for ill-defined organic pigments and fur colours.

I would feel much more comfortable if there was less of a focus on the pigment and the possibility of other preserved biomolecules was considered. If this was done then there would be a tangible link the extensive knowledge that exists in the field of organic geochemistry where the controls on lipid, carotenoid and porphyrin pigments, preservation and diagenetic pathways, including sulfurization, are extremely well-established. This would provide a more robust context for discussing the more challenging pigments subject of this paper.

Below we present the *reviewers' comments in italics* and our response in normal typeface. Where necessary, we also include direct references to the points in the manuscript that we have edited in order to deal with the reviewer's specific comment. Changes or additions to the original text are underlined.

Reviewer #1 (Remarks to the Author):

In the manuscript entitled "Pheomelanin pigment remnants mapped in fossils of extinct mammals" the authors (P.L. Manning, N.P. Edwards, U. Bergmann, J. Anné, W.I. Sellers, A. van Veelen, D. Sokaras, V. M. Egerton, R. Alonso-Mori, K. Ignatyev, B. van Dongen, K. Wakamatsu, S. Ito, F. Knoll and R.A. Wogelius) report on the development/validation of a chemical imaging methodology to map melanin pigments, and in the specific pheomelanins, in extant and fossil soft tissues.

This work is part of an up-to-date and very challenging research carried out by the authors and focused on the identification/localization of melanin pigments in extinct organisms, with the aim of adding important tiles for delineating the evolution pathways of some species.

The solid consciousness of the state of the art, the multidisciplinary expertise, the careful and meticulous approach and the innovative and sophisticated techniques employed to carry out this work, represent the real success of this paper.

The work is original and the topic is of interest for the audience of Nature Communications, the bibliography is accurate, the data are well presented and the conclusions are adequately discussed.

For these reasons I strongly recommend this paper for the publication on Nature Communications as is.

We thank the reviewer for these positive comments. There are no issues to address stemming from this review.

Reviewer #2 (Remarks to the Author):

The manuscript "Pheomelanin pigment remnants mapped..." submitted to Nature Communications is a comprehensive study on the trace metal distribution of a fossil mammal, suggesting a relationship with residual melanin pigments.

The result that the 3 million year old field mouse Apodemus was brown colored by pheomelanin and eumelanin similar than present-day Apodemus is not very surprising. However, the present study is interesting, because of the possible implications for the reconstruction of pigment patterns in extinct organisms in general.

The analytical work is very detailed and consistent. Although Zn elemental mapping of fossil specimens by SRS-XRF and XANES analysis as an approach to reveal the spatial distribution of melanins has already reported by ref. 7 (Barden et al., 2015), this study includes also the element sulfur. Limitations of the methodology would be that Zn occurs both in pheomelanin and eumelanin (although with a higher Zn/Cu ratio for pheomelanin) and other sulfur organic compounds may be present in fossils.

This is true, but one of our key previous findings in extant tissue was that Zn in pheomelanin is bonded directly to organic sulfur (the first reference in the manuscript: Edwards et al., 2016). In this new manuscript we show that this same diagnostic coordination chemistry is resolvable in an extinct mammalian species after 3 million years. This is the most important finding of this work, and gives palaeontology a definitive chemical method for distinguishing between eumelanin and pheomelanin residue. We have revised our explanation of this key finding to make sure it is absolutely clear to the reader:

Page 7, lines 22-27. “The mixed coordination environment for Zn most likely indicates the additional presence of residual eumelanin moieties, which is corroborated by the presence of eumelanin associated Cu and the correlation of Cu with Zn discussed below. However the clearly resolved spectroscopic details of Zn-organosulfur complexes (which do not occur in eumelanin) when combined with the mapped distributions of these compounds (which correlate with the residual integument of the organism) provides convincing evidence that the Zn-organosulfur residue is derived from a pheomelanin precursor.”

My major concern about this manuscript regards the reconstructed pigment distribution of Apodemus atavus. It should be considered that the depicted fossil specimens don't show the complete fossils. Generally, one part of fossils from Willershausen is preserved on a slab, while another part of the fossil remains on the counter slab. However, no information regarding the counter slabs of the two specimens is given in the manuscript. Both counter slab fossils (GZG.W.20027A and GZG.W.52-17393a) have been reported in the original description of the Apodemus specimens from Willershausen (S. Rietschel, G. Storch. Senckenbergiana Lethaea 54, 491-519 (1974). This paper should be cited in the main article, not in the supplementary part.

Page 4, line 6: This is completely reasonable and we have moved this reference into the main article, see current reference #27.

Moreover, some material may have been removed during former preparation of the fossils. It is obvious that some parts of the tissue surface are lost or not considered in the elementary mapping (especially in the specimen depicted in Fig. 1), because otherwise also the bones of the fossil would be covered with tissue material and fur. In the specimen depicted in Fig. 1 no tissue material and fur is preserved in the tail (only bones!) and most parts of the feet and the jaw, thus no statement regarding the pigmentation of these parts is possible (last paragraph of page 9, and life reconstruction). Because of this partial loss of material on the analyzed slabs, the layer and thickness of the preserved tissue material and fur is not the same in different parts of the fossil. In my view, this explains to a significant extent the uneven distribution of Zn, Cu, and S, especially of the specimen depicted in Fig. 1. Although theoretically resolving of pigment patterns may be possible for other fossils, contrary to the authors, I see no evidence for pigment patterning of Apodemus atavus.

We thank the reviewer for these comments. He or she is absolutely correct in questioning our assertions with respect to patterning because of possible uneven separation between part and counterpart. We did not have access to the counterparts for scanning and did not have photographs when we submitted the manuscript for publication, and the fact that we did not consider this possibility was an oversight on our part. We have taken this seriously and obtained high resolution photographs of the counterparts of both fossils. Below we present a series of images of part and counterpart in order to respond to these comments concerning whether our chemical images can be explained by patterning as opposed to uneven separation.

Figure S12A presents a photo montage of the lateral fossil [GZG.W.20027, b, a(1), a(2)]. Scale bar is 1 cm. The counterpart is incomplete and in two fragments. However, the regions corresponding to the sinistral fore and hind limbs (I and II respectively) as well as the base of the tail (III) are preserved and may be compared. Figure S12B presents magnified views of these regions (scale bars 0.5 cm). Region I, corresponding to the forelimb, shows that uneven separation along the bedding plane cannot account for the lack of pigment residue in this region; indeed more fossil material is hosted within the part that we imaged as opposed to the counterpart (arrows indicate

residue). Exactly the same result is found by examining regions II and III, the hind limb and base of tail: more fossil material is present on the scanned part, hence if any pigment residue were present it would most likely be found on the main part of the fossil which we scanned. We also show part and counterpart of the dorsal fossil [GZG.W.17393 a and b] in Figure S12C. Due to the orientation of this fossil, only the hind limb (region IV) may be resolved for comparison. However, here again uneven separation cannot explain the lack of pigment residue; there is no evidence that soft tissue remnants are preferentially deposited on the counterpart (Figure S12D shows the detail). Therefore in all four areas where part and counterpart can be compared, our conclusion that the absence of Zn- and S-bearing organic compounds in these regions is due to decreased amounts of pigment residue (i.e. patterning) is unequivocally confirmed. Unfortunately, the ear and jaw regions in the counterpart of the lateral fossil are not present and so we cannot be certain that the ears and jaw were lighter relative to the body. We have amended our conclusions with respect to patterning and thank the reviewer for noting this point:

Page 11: lines 23-28: “We have examined photographs of the counterparts of both specimens (see Supp. Inf. Figure S12) and the lack of either organosulfur-Zn compounds or Cu in the feet and tail cannot be explained by uneven separation along the bedding plane and thus indicates that these areas were almost certainly not pigmented, as is also the case with the extant species *A. flavicollis*. Unfortunately the ears and jaw region of the lateral fossil are not preserved on the counterpart, and so we can only conclude that the ears and lower jaw region of *A. atavus* were probably not pigmented.”

We have furthermore added the following series of images to the Supp. Inf. File as Figure S12.

Figure S12A. Optical photograph of the lateral fossil [GZG.W.20027b, left] compared to its counterparts [GZG.W.20027a(1), bottom right; GZG.W.20027a(2), top right]. Scale bar = 1 cm. Three regions are highlighted for comparison to check whether uneven separation of soft tissue between part and counterpart could explain the lack of chemical residue in the scanned images.

Figure S12B. Detailed photographic images of the scanned regions of the lateral fossil [GZG.W.20027b] presented on the left compared to its counterpart regions on the right [GZG.W.20027a(1), region 1; GZG.W.20027a(2), regions II and III]. Scale bars = 0.5 cm. Arrows indicate expected locations of soft tissue. There is no evidence of uneven separation causing more soft tissue to be preserved on the counterpart in any of these three regions. Indeed, in all three cases more of the fossil residue appears on the left. Therefore the lack of pigment residue within these regions is most probably due to low levels in the original organism.

Figure S12C. Optical photograph of the dorsal fossil [GZG.W.17393a, left] compared to its counterpart [GZG.W.17393b, right]. Scale bar in cm. A fourth region is highlighted for comparison to check whether uneven separation of soft tissue between part and counterpart could explain the lack of chemical residue in the scanned images.

Figure S12D. Detailed photographic images of the scanned region IV of the dorsal fossil [GZG.W.17393a] presented on the left compared to its counterpart region on the right [GZG.W.17393b]. Scale bar = 0.5 cm. Arrows indicate expected locations of soft tissue. There is no evidence of uneven separation causing more soft tissue to be preserved on the counterpart. Therefore the lack of pigment residue within this region is also most probably due to low levels in the original organism.

Reviewer #3 (Remarks to the Author):

Reviewer #3 makes some strongly positive comments for which we are grateful: “Overall, the paper contains some wonderful images, (and) great spectroscopy...”

The reviewer is however sceptical concerning our conclusion that the Zn-organosulfur compounds we identify within the fossil integument represent pheomelanin residue. Indeed, the reviewer asserts that identifying pigments in fossils is an “*insurmountable conundrum*,” because pigments are complex and they degrade during fossilisation. The reviewer furthermore has difficulty seeing how sulfur metal complexes formed during fossilisation can “*be distinguished from endogenous S metal complexes.*” These are important concerns that are critical to this work and we respond to these main points and other comments raised by the reviewer below in detail within our point-by-point rebuttal. Based on those comments we have taken the opportunity to revise, clarify, and strengthen the manuscript with respect to why we conclude that the Zn-organosulfur species are endogenous and why they are most probably the residue of pheomelanin breakdown. We have sought to clarify the text to accommodate all of the reviewer’s comments as comprehensively as possible, but we have restricted these amendments because while we understand that a *Nature Communications* manuscript needs to be more generally comprehensible than a subject specific journal, we are also aware that there are limits to how far we can explain things given the word count restrictions. Given this, we have edited the text where appropriate and as directed by the Editor we have moved all of our analytical results from the Supp. Inf. section into the main text to ensure that the full scope of our study is more coherently presented.

Now we turn to our point-by-point rebuttal.

I. The major claim of the paper is that the authors have identified pigments in fossil mammals. They are careful in the title of the paper to avoid that claim that they have identified the actual pigments, rather they have identified "remnants" of the pigments. However, the language of the paper very quickly reports the finding of pigments often without caveats.

We tried to be very careful to avoid exactly this potential misunderstanding. Therefore, the only time we refer explicitly to actual intact pigments in this manuscript is when we refer to extant standards which contain pigments as confirmed by AHPO-HPLC analysis. Everywhere else in the manuscript we refer to the fossil organic compounds as either “residue” or “derived”. However, we can see that because we present results from extant standards as well as from fossil material, it is possible that some confusion may have arisen. We have therefore reorganized and edited sections which discuss both sets of results so that there is no ambiguity.

Page 8, lines 13-29:

“Both extant murid specimens have higher transition element concentrations in darker pigmented fur. Cu, when associated with organic compounds, has been shown to be an indicator of eumelanin distribution². However, it is clear that the main trace metal difference in the extant *A. sylvaticus* between light and dark regions is the Zn/Cu ratio, which is much higher in the pigmented, pheomelanin-rich region. This is consistent with previous data on extant integument¹ and with the Zn XAS and XRF data for the extant material presented here, both of which resolve an affinity between Zn and pheomelanin-associated organic sulfur. For example, Supp. Inf. Figure S7 shows analyses of red human hair with Zn dominating the metal loading. (Supp. Inf. Figure S8 presents details of additional XRF point analyses.) If we then compare the results from the extant tissue to the fossils, the high Zn/Cu ratios in the integuments of both fossil *A. atavus* specimens (Table 2), plus the similarities in the Zn-EXAFS of the fossils to known pheomelanin rich tissue (Table 1), plus the relatively good correlations between organic S, Zn, and Cu (Figure S2), all point towards the organosulfur-Zn compounds within the fossil being derived from a pheomelanin precursor. Furthermore, the correlation between Cu and Zn presented in Supp. Inf. Figure S2C and D combined with the mixed

first shell coordination environment for Zn suggests that the fossil residue is derived from a mixture of both original eumelanin and pheomelanin, implying that *A. atavus* would be best described as a “mixed” melanic phenotype.”

Page 9, lines 3-8:

“Thus even if a large fraction of these benzothiazole units were destroyed by aging there would still be sufficient S binding sites to produce the mixed Zn coordination environment observed in the *A. atavus* fossil specimens. Therefore the trace metal point analyses and melanin pigment assay are consistent with the mapping and spectroscopy, indicating a relationship between sulfur bonded Zn and pheomelanin in the extant tissue as well as confirming the presence of residue in the fossil that is most probably derived from both types of melanin pigments.”

We again note that we have clarified our key finding in response to reviewer #2 above which should also help make it absolutely clear that what we find here is residue:

Page 7, lines 22-27. “The mixed coordination environment for Zn most likely indicates the additional presence of residual eumelanin moieties, which is corroborated by the presence of eumelanin associated Cu and the correlation of Cu with Zn discussed below. However the clearly resolved spectroscopic details of Zn-organosulfur complexes (which do not occur in eumelanin) when combined with the mapped distributions of these compounds (which correlate with the residual integument of the organism) provides convincing evidence that the Zn-organosulfur residue is derived from a pheomelanin precursor.”

II. For me this is where the major challenge of this field begins and ends. The organic pigments they refer to are pheomelanins, which as I understand are related to melanins, which are notoriously difficult to define structurally even in modern tissues. Thus, it is not the organic constituents that are identified but inorganic elements that are claimed to be specifically associated with these pigments. The claims of the paper rests on the purported specificity of these associations. The introduction provides a long summary of the field and critiques other studies drawing attention to the limitations of the TOF-SIMS method, then proposes a new approach based on chemical imaging methodology. They say this "show[s] that organosulfur-Zn complexes are indicators of pheomelanin (red pigment)". I looked back at several of the cited papers which confirmed that at no point are the organic pigments ever structurally unambiguously identified.

Eumelanin is a group of heterogeneous biopolymers arising from tyrosinase-catalyzed oxidation of L-tyrosine without involvement of L-cysteine in this process.

In our opinion, it is not meaningful to discuss its full structure. It is only meaningful to discuss:

- 1) the process of polymerization of the monomeric precursors, DHI and DHICA to form melanin polymers,
- 2) aggregation of the melanin polymers to form the supramolecular structure, and
- 3) precise chemical (and physical) characterization of the melanin polymers including the ratio of DHI to DHICA (such as our co-authors Ito and Wakamatsu are performing). For a review, see: The Supramolecular Buildup of Eumelanin: Structures, Mechanisms, Controllability. Büngeler A, Hämisch B, Strube OI, Int J Mol Sci. 2017 Sep 6;18(9). pii: E1901.

As regards pheomelanin, it is also a group of heterogeneous biopolymers arising from tyrosinase-catalyzed oxidation of L-tyrosine with the intervention of L-cysteine. In this case it is only meaningful to discuss:

- 1) the process of the monomeric precursors, cysteinyl dopas, to form pheomelanin polymers, and
- 2) precise chemical (and physical) characterization of the melanin polymers including the ratio of benzothiazine to benzothiazole units. Our recent study aiming at a more specific characterization of eumelanin and pheomelanin was recently published (Pigment Cell Melanoma Res. 31, 393-403, 2018. DOI: 10.1111/pcmr. 12673.)

Therefore the full structural determination of either eumelanin or pheomelanin, as discussed above, is not meaningful. In the case of pheomelanin, the benzothiazole-benzothiazine markers are diagnostic of pheomelanin and one does not need to identify the entire molecular structure of pheomelanin in order to demonstrate the presence of this pigment; one would not (indeed could not) do this even in extant tissue and the full molecule would certainly not be expected to survive over geological time in fossil material in any case.

Indeed, we explain this in detail in the beginning of the manuscript, as we hypothesize that the relative instability of pheomelanin is the key reason why it has been difficult to identify using other more traditional methods. In this paragraph we now allude to method sensitivity:

Page 3, lines 15 and following:

“This final point is crucial with respect to detecting pheomelanin residue. Their (Colleary et al. 2015) principal component analysis strongly implies that the collapse in their second principal component (PC2) as a function of age (or degradation) is most likely a direct function of sulfur loss, as the “sulfurous” smell of their experimentally reacted pheomelanin corroborates. Pure eumelanin does not contain sulfur, and so the documented collapse in PC2 in their work most likely explains why the pheomelanin biomarkers are not detectable in the ToF-SIMS analyses: the pheomelanin biomarkers become undetectable relative to the eumelanin biomarkers using their method as sulfur compounds are devolatilized from the organic residue.”

However, what our current manuscript highlights is that the identification of molecular fragments and metal coordination complexes which are identical to fragments and complexes within extant pigmented tissue is a powerful chemical or “structural” indicator that the residual material is potentially derived from pigmented precursors. Using AHPO-HPLC methodology which is the gold standard for analysis of extant tissue, we have so far failed to detect pheomelanin in various fossil samples. Particularly important for work on fossil tissue was our finding (Pigment Cell Melanoma Res. 31, 393-403, 2018. DOI: 10.1111/pcmr. 12673) that pheomelanin in dark human hair (or any tissue samples) is a tough molecule surviving heating in 6 M HCl at 110°C for 16 h. Therefore even though the entire benzothiazole inventory in pheomelanin will probably not survive through an extended process of diagenesis, sensitive spatially resolved methods may be able to detect trace residue bonded to metals. These analytical results and the work of others are in fact the driving force for this work: to see whether the organic-metal complexes which we have characterized as pheomelanin biomarkers in extant integument can be resolved in relatively young fossils and thereby provide a more robust probe for fossil pheomelanin as we say in the introduction:

Page 3, line 40 and following: “Therefore the work presented here seeks to test whether non-destructive and extremely sensitive synchrotron-based methods (including XAS) may be able to resolve traces of pheomelanin residue in fossil specimens...”

We also add for comparison that our study of eumelanin residue in fossil integument (Wogelius et al. 2011) cited the most extensive structural computational work on that pigment (Kaxiras et al., 2006; doi:10.1103/PhysRevLett.97.218102 pmid:17155775), and showed by direct comparison that the Cu coordination chemistry or “structure” in the fossil soft tissue residue was identical to the Cu coordination chemistry or “structure” of the lowest energy configuration of porphyrin rings in eumelanin. Taken together with all of our additional analyses of that fossil as well as comparative analyses of extant specimens that work provided the first ever chemically based detection of eumelanin pigment residue within fossils. It has been *post facto* corroborated by a number of studies using XAS and other methodologies. This is exactly what the current work has sought to do with respect to the more difficult problem of resolving pheomelanin pigment residue: however it is biomarkers, or clusters, or fragments, that will tell the story, not fully intact melanin molecules.

III. The presence of pigments are always inferred from the distributions of various inorganic elements, which the authors have found to be associated with pigments in modern fur and feathers. Given the

latter they authors then use the occurrence of these inorganic elements to map the occurrence of different coloured pigments on fossils - this is quite a leap of faith given the complexities of fossilisation systems.

We do not here, and never have, identified pigment or any organic residue simply through elemental mapping. The relationship between eumelanin pigmentation and organically bound trace metals has been documented previously, not only in our work, as we explain and reference early in the manuscript:

Page 2, line 13-15 (emphasis added):

“Because Cu is the metal co-factor in the enzymatic process forming eumelanin¹⁸, elevated concentrations of **organically bound Cu** can typically be correlated with eumelanin-rich tissue^{2,19,20}.”

Furthermore, the current manuscript includes extensive and detailed spectroscopic analysis of the coordination chemistry and oxidation states of multiple elements. We believe this is clearly stated:

Page 6, line 3:

“**heterocyclic S** is resolvable within the fossil soft tissue regions”

Page 7, lines 10-13:

“...XAS analysis shows that **Zn-organosulfur** compounds are unquestionably present within the fossil fur regions, and furthermore shows that these complexes have a coordination chemistry for Zn that is directly comparable to the Zn coordination environment in pheomelanin pigmented hair and fur.”

Page 7, lines 19-21:

“Therefore, by first confirming via spectroscopy that the **Zn has a biochemical affinity**, the **combined maps of Zn plus organosulfur** may then be used to reconstruct pheomelanin pigment distribution in these fossils as previously shown for extant feathers¹.”

These statements clearly indicate that it is not simply element mapping, but the combination of spectroscopy with spatial information that implies the distribution of pheomelanin residue. We apologize if this was not clear enough, and our response to the first comment of reviewer #2 also addresses this point (page 7, lines 22-27).

We feel obliged to also reply that the synchrotron work was presented along with SEM work, FTIR analysis, diffraction, and optical methods. We used multiple lines of evidence besides elemental mapping and X-ray spectroscopy to distinguish endogenous from exogenous compounds. It is possible that because we present extant standards along with fossil specimens, as well as the fact that to save space we put some of our analytical work in the Supplementary Information section, that the reviewer did not clearly see the full line of evidence. If that was the case we again apologize for our lack of clarity.

We have edited the manuscript significantly to address this:

1) The inorganic sulfate-bearing mineral gypsum is clearly resolved on the bedding plane via XRD. This is discriminated from the organic sulfur in the fossil later via specific oxidation state mapping of sulfur.

Page 4, lines 16-17:

“Glancing incidence XRD analysis identified the minerals gypsum, calcite, quartz, ankerite, and muscovite in the sedimentary matrix plus hydroxyapatite in the fossil bone (Figure S3).”

2) FTIR spectroscopy was completed on the fossil integument:

Page 9, lines 11-25:

“FTIR spectroscopy (Figure S10) was also completed in order to further test the endogeneity of the chemistry of the fossil soft tissue. The fossil soft tissue FTIR spectra are completely different from the background matrix (see Figure S10), containing clear strong organic absorption bands including: amide I, II, and III, hydrocarbon, and C=O. The matrix is however dominated by carbonate and silicate absorption bands and includes only a single weak absorption band for trace organics (C=O). In the fossil, the amide bands would be expected as a breakdown product of alpha keratin from both skin and fur, and the hydrocarbons originate from the lipid and other aliphatic components in the original organism. The C=O moiety is consistent with the presence of fatty acids derived from the original biochemistry and may provide some of the oxygen terminated functional groups which we see via EXAFS bonded to the trace metals. We also note that original pheomelanin is rich in carboxylic functional groups due to the presence of abundant arginine and therefore at least part of the C=O groups may also be derived from the pheomelanin pigment itself. These results clearly show the presence of amide groups within the fossil, thus resolving organic residue that is indicative of integument material and is consistent with the X-ray imaging and spectroscopy, as also shown in several previous studies of extant and exceptionally preserved fossil integument (e.g. ^{2,13,14,33}).”

3) ESEM was completed on the fossil surface:

Page 9, line 26 and following:

Environmental Scanning Electron Microscopy (ESEM) was applied in order to determine whether melanosomal micro-structures might be present within the integument residue. Other studies have based the reconstruction of pigment distribution on putative fossilized melanosomic structures, with rod-shaped structures approximately 0.5 to 2 μm in length classified as fossil eumelanosomes and spheroidal structures with a diameter of approximately 1 μm classified as fossil pheomelanosomes ^{22,34-37}. Doubts about the interpretation of these structures have been raised due to their similarity to microbial fossil products²⁵. The structural approach also has limitations due to the fact that it relies on electron microscopy and therefore only examines a tiny fraction of the surface area of fossil material requiring extrapolation to whole organism colour reconstruction. We furthermore note that in all but the most extreme cases, both types of melanosome will be found together: see Table 2 for the proportions of melanin pigments in extant brown, blonde, and red human hair. Even the light blonde hair has over 30% eumelanin pigment, and so simply resolving eumelanosomes or pheomelanosomes in a fossil specimen of hair such as this would not unequivocally reveal specimen pigment predominance.

ESEM revealed the presence of numerous spheroidal bodies (0.8 to 2 μm ; see Figure S11), calcite rhombs, phyllosilicate platelets, abraded sediment grains (quartz and feldspar), and Ti-rich aquatic microbodies (chrysophyte cysts: ~30 micron diameter). ESEM energy dispersive spectrometry (EDS) revealed high Ca concentrations in the matrix, consistent with the XRF analyses, the XRD results, and the presence of calcite rhombs. These abundant micron-sized spheroidal bodies blanket much of the fur and skin regions of the fossil *A. atavus* (see Table S3 and Figure S10). The corrugated surfaces, shapes, and size range of these are consistent with fossilized melanosomes. Destructive analysis of these precious specimens was not allowed, therefore we could not scrape away the bodies to determine whether they are penetrative or to analyse them in isolation from the underlying material. Thus we simply point out that these spheroidal bodies do indeed correlate with the fossil and are absent from the matrix bedding plane, hence they are definitely associated with the fossil.

Similar microbodies were found to correlate with organosulfur-Zn complexes in a postulated pheomelanin pigmented amphibian⁷. However it should be made clear that the Zn spectroscopy and fluorescence imaging presented here would not be compromised if these micro-bodies were reclassified as microbes via some other analytical approach at a later date, because the Zn-EXAFS is determined by fluorescence with an attenuation depth of approximately 40 microns. Therefore any surface film of micro-bodies would contribute far less than 5% of the fluoresced Zn x-rays and most likely would not significantly contribute to the signal from the underlying mouse fossil. In any case, the presence of these micron-sized spheroidal bodies which dominate the exposed fur and skin regions

of the fossil *A. atavus* (see Supp. Inf. Figure S11 A-D) suggests that fossil pheomelanosomes may be present which further supports the chemical evidence for the presence of pheomelanin residue.”

4) Microfocus XRF was completed to complement SRS-XRF images:

Pge 10, lines 13-25:

“Finally, in order to improve the spatial resolution of our X-ray imaging so that we could be certain that the distribution of residual chemistry displayed biological structural control, we completed microfocus XRF mapping (Supp. Inf. Fig. S11). This showed that the Zn distribution resolved hair-like filaments within the *A. atavus* specimens and that lineations on the periphery grade into masses of higher Zn concentration from distal to proximate locations, as would be expected if this metal pattern has been derived from original fur. Our results therefore go beyond the microstructures and allow a broad spectrum of the chemical inventory of the integument to be used to reconstruct the original distribution of organismal biochemistry. We have also mapped the elemental inventories of two other specimens sampled from the Willershausen locality, including an intact single feather and a partial, but articulated, frog. Patterns in both of these fossils related to soft tissue chemistry can be resolved, however neither of these fossils displays the same type of correlation between zinc and organic sulfur as shown by the *A. atavus* specimens. This supports the hypothesis that the patterning seen in these mouse fossils is endogenous and is not a generic feature of the taphonomy at Willershausen.”

IV. One paper I looked at used various inorganic elements to map the pigments a 150 million year old bird feather. The curious thing about the images they showed is that the visible light image actually appeared to show a clearer representation of the feather than the element maps. I guess the authors would argue that the blurring in the element images is caused by some sort of diffusion during fossilisation - this made me feel uncomfortable - perhaps the elements arose diagenetically by diffusion.

There are four key reasons for “blurring” in elemental X-ray maps relative to photographs.

First of all, the organic residue that gives high optical contrast relative to the sedimentary matrix is not uniform in chemistry, just as the original tissue is not completely homogeneous. Therefore, some parts of a feather or hair may be higher in Zn, Cu, or Ca while the distribution of C is more uniform. Then, when fossilized, the Cu, Zn, or Ca rich regions when imaged by X-rays may look “blurry” relative to the photographic image which depends mostly on contrast generated by C content.

Secondly, the X-ray images are often produced from elements present at trace concentrations, in our case we can image as low as 5 ppm. However, at these low concentrations when we are scanning at a rate of milliseconds per pixel the images may become slightly blurred due to low count rates. This is due to counting statistics.

Thirdly, the pixel size on a digital camera ranges between approximately 1 to 10 microns. For rapid scanning, our beam size (and pixel resolution), as we state in the manuscript, was either 50 or 100 microns. Hence in order to scan rapidly we sacrifice some spatial resolution. This is why we also completed microfocus mapping which, again as we say in the manuscript, gives us near micron resolution. Figure S11 presents this data which has a pixel size of 2 x 5 microns and clearly resolves the radial filament pattern of fur.

Finally, the emission depth of various characteristic energy X-ray photons is highly variable as one goes from S to Zn. Sulfur, at a fluorescence emission energy of 2.3 keV, has an attenuation depth in a typical sedimentary matrix of approximately 8 microns, while Zn at 8.6 keV has an attenuation depth of approximately 40 microns. Hence there will be some “blurring” of different elements relative to each other based on variable emission depths within the specimen- in other words there is a three-dimensional aspect to X-ray fluorescence images which is not the case with visible light photographic imaging which can be approximated much more as a 2-D phenomena.

V. The paper is very long and extremely complex in its explanation of the various spectroscopic techniques used wrapped up with the detailed morphological descriptions. For this paper to be publishable I feel it needs to be shortened and the language simplified. I was left asking why the explanations were so complex. I am suspecting as this is a controversial area which is clearly energising a number of groups and the rivalries are quite strong and so the high level of detail is included to head off criticism.

The new draft has many revisions, and we believe that the level of detail we provide is required to address the complexity of the chemistry involved. We are happy to work with the editor to simplify any language if needed.

VI. The basic problem as I see it is that the various workers in this field are wrestling with a seemingly insurmountable conundrum - the pigments are difficult to define in modern animals and identification in fossils made all the worse by fossilisation, thus rather than identify the pigments they use non-specific tracers which might in themselves be impacted upon by fossilisation. I'm not sure I see how they can break out of this.

This concern is repeated later in the review:

And if the right elements are present then S metal complexes would be entirely expected but how can these be distinguished from endogenous S metal complexes - this would seem to be another conundrum?

As we discuss in detail above, there is no need to unambiguously resolve the entire molecular structure of pheomelanin or eumelanin within a tissue in order to prove the presence of either pigment or its residue. Molecular fragments can be diagnostic of specific compounds- the field of biochemistry is full of examples of this principle that go well beyond pigment research. This principle is also well-established in organic geochemistry.

For example the carotenoid fucoxanthin undergoes rapid degradation in sediments, ultimately leading to the much smaller loliolide, indicating that the latter can be used as a biomarker for the presence of fucoxanthin (Repeta 1989, [https://doi.org/10.1016/0016-7037\(89\)90012-4](https://doi.org/10.1016/0016-7037(89)90012-4); Sinnighe Damsté and Koopmans, 1997, *Pure and Applied Chemistry* 69, 2067-2074). In addition, if you want to determine if green sulfur bacteria were abundantly present in the water column during the time of deposition (to determine the presence/absence of photic zone euxinia) one often cannot detect the original diagnostic carotenoid isorenieratene anymore and has to rely on the quantification of a whole range of (smaller) diagnostic diagenesis products (van Dongen et al. 2006, doi.org/10.1016/j.orggeochem.2006.05.007). Furthermore, rather than directly identifying lignin present in sediments tracer moieties obtained by pyrolysis GC/MS are typically used to determine the relative abundance of lignin (Sparkes et al. 2016, <https://doi.org/10.5194/tc-10-2485-2016>).

Our use of metal coordination complexes as a molecular biomarker grows out of this approach. While we agree that, of course, there are possibilities of contamination or other breakdown pathways which could also produce the organic compounds we detect, we also presented a significant quantity of additional data including spatial distributions (mapped biological structures), concentrations (quantitative amounts that match extant concentrations), FTIR analysis (C=O functional group distributions correspond to organic S and Zn distributions), and ESEM structural identification of pheomelanosome-shaped bodies (see above). While the questions this reviewer raises concerning endogeneity are *prima facie* reasonable, the reviewer seems to have missed the detailed information concerning endogeneity presented within the section of the manuscript sub-headed “*Trace Metals, Melanin Assay, and Other Analyses*” (see our response above to point III) as well as the paragraph in the Supplementary Information titled “**Endogeneity**”. Because the reviewer seems to have missed these important points, we have also moved the “endogeneity” paragraph into the main text in order to explicitly deal with the issue of endogeneity.

Page 10, lines 26-43:

“For specimens that are ~3 million years old we cannot *a priori* dismiss the possibility that geochemical processes may have altered the distribution of Zn and S. However our results indicate that for these specimens this is unlikely, given that Zn and organosulfur compounds map discretely within the fur and any postulated geochemical process would have to deposit organosulfur compounds of Zn in such a way as to mimic the original organism’s structure. All of the sulfur concentration measurements via synchrotron XRF and ESEM-EDS (Figure S8, Table S1, Table S3) have an average of 5.2% (range 1.14 to 7.75, n = 9) comparable to the sulfur content measured for extant mouse fur (5.51%, Table S1) and therefore there is no requirement for mass transfer of sulfur into this system to account for the observed sulfur content. Sulfurization has been considered, but given that there is no sulfur enrichment in the tissue analysed in our study and no such sulfurization determined for other Willershausen specimens³⁸ it is highly improbable that sulfurization is responsible for the distribution of organic sulfur in our two specimens. Likewise, the range of melanin-associated trace metal concentrations within the fossil fur (21 – 2480 ppm) is comparable to the range exhibited within extant hair and fur (10 – 2163 ppm) and therefore an influx of metals via geochemical fluids into the system is not required to account for the chemistry we resolve. Small Ti-rich cysts are present, but these are discrete structures which are easily discriminated from the residual fur. Furthermore, the precipitation of inorganic Zn minerals, the most likely process to introduce Zn into the fossilized integument, is wholly inconsistent with the XAS spectroscopy.”

The reviewer comments further that: “*Arguments related to the so-called fossil organic S also become extremely circular.*” First of all, the sulfur spectroscopy definitively shows that the sulfur we are imaging within the integument is dominated by organic sulfur species; it is unequivocally fossil organic S.

Page 5, line 5 and following (emphasis added in boldface):

“These results are reproduced with the dorsal fossil specimen, and a direct comparison of Fig. 2A, B, and C with D, E, and F shows that exactly the same result is obtained from both specimens: sulfate is present on the bedding plane (Fig. 2F) and the **reduced organic species of sulfur are concentrated in the soft tissue residue** (Fig. 2E). Indeed, this allows us to spatially resolve the radiating texture of hair along with patches of skin remnants (Fig. 2. B & E). Figure 2G is a false colour composite image which unequivocally shows that **Zn and organic sulfur are correlated within the integument** of the dorsal fossil just as in the lateral specimen. (Supplementary maps of the dorsal fossil are given in Supp. Inf. Figure S4A and B).”

Secondly, our finding that the sulfur concentrations (by weight) in the fossil fur are almost identical to the sulfur concentrations in modern tissue is not a circular argument, it is a direct measurement that definitively shows that the addition of sulfur to the fossil via sulfurization is not required and is, on the contrary, unlikely.

VII. This is followed by the comment “*Sulfurization of organic matter is a well know phenomenon in diagenesis (sic) and thus the extensive presence of S is to be expected.*”

There is no documentation of sulfurization at Willershausen (see our reference #38). We now discuss this explicitly in the endogeneity paragraph in order to respond to this criticism.

Page 10, lines 34-37:

“Sulfurization has been considered, but given that there is no sulfur enrichment in the tissue analysed in our study and no such sulfurization determined for other Willershausen specimens³⁸ it is highly improbable that sulfurization is responsible for the distribution of organic sulfur in our two specimens.”

VIII. Overall, the paper contains some wonderful images, great spectroscopy but there is considerable circulatory in the arguments made for the use of non specific inorganic elements as proxies for ill-defined organic pigments and fur colours.

Zn here is bound to organosulfur compounds and is clearly not a “*non specific inorganic element*.” This is a fundamentally important point: metals form complexes with organic molecules and the coordination chemistry may be diagnostic of that particular metallome. There are two sections of our manuscript that provide detailed chemical information concerning the organic nature of the S and Zn which we image, and show that the Zn has inner-sphere S coordinated to it exactly as in pheomelanin.

Furthermore, the evidence we present does not agree with the reviewer’s comment concerning “*ill-defined organic pigments*”. We have provided 12 current references to the structure of eumelanin and pheomelanin, and we are extremely clear about the diagnostic chemical indicators for pheomelanin. We cite three previously published studies from other research groups which resolve pigment residue from molecular fragments or clusters (our references 20, 22, 23). This is well-established scientific fact, and we have contributed to this literature with previous studies of both eumelanin and pheomelanin and perhaps most pertinently by being the first investigators to unambiguously resolve the benzo-sulfur moiety in pheomelanin from extant species via X-ray absorption spectroscopy (Edwards et al., 2016). We apply this methodology here and produce results consistent with pheomelanin residue. We wholeheartedly agree with this reviewer with respect to the burden of proof concerning endogeneity. Indeed, the entire first section of our methods paper (Bergmann et al., 2012: *Annual Review of Analytical Chemistry* **5**, 361-389) deals with the challenges involved in analyzing ancient material. **Precisely because we are extremely concerned about endogeneity**, we go beyond sulfur spectroscopy and beyond elemental mapping in this manuscript and complete Zn X-ray absorption spectroscopy along with several other analytical tests all of which unequivocally corroborate our initial sulfur spectroscopy data. We do not believe it is fair to refer to our findings as “ill-defined” or “non specific”.

Please also see the sub-sections titled “Sulfur X-ray Spectroscopy” and “Zinc X-ray Spectroscopy”.

IX. I would feel much more comfortable if there was less of a focus on the pigment and the possibility of other preserved biomolecules was considered. If this was done then there would be a tangible link the extensive knowledge that exists in the field of organic geochemistry where the controls on lipid, carotenoid and porphyrin pigments, preservation and diagenetic pathways, including sulfurization, are extremely well-established. This would provide a more robust context for discussing the more challenging pigments subject of this paper.

Unfortunately, full organic analysis of this fossil is not possible because it would require complete destruction of these precious fossils. We know from direct, extremely distressing experience, that to acquire sufficient quantities of material for pyrolysis-GC/MS analysis from fossils such as these that we would need to scrape off nearly the entire thin film of bedding plane fossil. We have in fact done near complete bedding-plane fossil destruction previously with a less precious fossil in order to prove that such destruction is not necessary: see (Edwards et al., 2011; DOI: 10.1098/rspb.2011.0135). In that case, we learned that synchrotron and infra-red methods are more suitable to study trace organic sulfur chemistry because the column signal for trace sulfur-bearing compounds is swamped by the convolved masses of carbon compounds. Because we are focussing on sulfur compounds and transition metal coordination chemistry, the X-ray and FTIR methods are the best approach. In any case, the museum curator would not permit us to perform destructive analysis of such rare specimens.

Therefore, in the present manuscript, because of the unique and precious nature of this fossil we are limited to non-destructive methods, which is one of the main motivations we have for developing X-ray and infra-red methods. We provide the “link” to organic geochemistry by completing non-destructive FTIR spectroscopy and imaging as discussed above. The clear presence of amide groups is

consistent with integument residue (structural protein remnants from keratin) as we have shown in multiple previous publications referenced in this manuscript:

2. Wogelius R. A., Manning P. L., Barden H. E., Edwards N. P., Webb S. M., Sellers W. I., Taylor K. G., Larson P. L., Dodson P., You H., Da-qing L., and Bergmann U., (2011) "Trace metals as biomarkers for eumelanin pigment in the fossil record," *Science* 333, 1622-1626.

13. Barden H.E., Wogelius R.A., Li D., Manning P.L., Edwards N.P., and van Dongen B.E. (2011) "Morphological and geochemical evidence of eumelanin preservation in the feathers of the Early Cretaceous bird, *Gansus yumenensis*," *PLoS ONE* 6(10): e25494.

14. Edwards N. P., Barden H. E., van Dongen B. E., Manning P. L., Larson P. L., Bergmann U., Sellers W. I., and Wogelius R. A., (2011) "Infrared mapping resolves soft tissue preservation in 50 million year-old reptile skin," *Proc. Roy. Soc. B: Biol. Sci.* 278, 3209-3218.

33. Manning P.L., Morris P.M., McMahon A., Jones E., Gize A., Macquaker J.H.S, Wolff G., Marshall J., Taylor K.G., Lyson T., Gaskell S., Reamtong O., Sellers W.I, van Dongen B.E., Buckley M. and Wogelius R.A. (2009) "Mineralized Soft-Tissue Structure and Chemistry in a Mummified Hadrosaur from the Hell Creek Formation, North Dakota (USA)," *Proc. Roy. Soc. B: Biol. Sci.* 276, 3429-3437.

We show that high quantities of C=O functional groups are present within the residual integument but not the sedimentary matrix, consistent with fatty acids derived from the fossil organism. This further supports the endogeneity of the organic material we resolve by synchrotron and electron beam methods. We provide abundant information linking this work into the field of organic geochemistry, even to the point of explaining the breakdown pathway for pheomelanin (Supp. Note III) and constraining the requirements for the clearly observed exceptional preservation seen in these fossils. This is all present within the Supp. Inf. as well as explicitly covered in our discussion of the results presented on Figure 3. We initially did not put all of the chemical details in the main article for reasons of length.

In order to deal with the reviewer's comments we have added some clarifying text highlighting the fact that we have completed corroborating analyses:

Page 4, lines 1-4:

"In order to verify our results we have completed additional analyses including state-of-the-art destructive AHPO-HPLC measurements on standards such that the synchrotron results with fossils may be directly compared to the pigment loadings in extant organisms."

We have also moved five paragraphs from the Supp. Inf. to the main text, see response to points III and VI above.

X. The reviewer then lists several other biomolecules that we should consider. His first compound is "*lipids*." Please see the FTIR paragraph because we clearly state that the hydrocarbons identified via FTIR are most likely lipid associated. Concerning "*carotenoids*"; mammals do not synthesize carotenoids, and therefore any such signal in a mouse fossil would be overwhelmed by melanin residue. Finally, "*porphyrin pigments*"; eumelanin is a "porphyrin pigment". One of our key findings is that Zn is not in the centre of a porphyrin ring, and so we have in fact already considered the most pertinent "porphyrin" pigment. The distribution of Cu and Zn, and the Zn coordination chemistry are consistent with some quantity of residual eumelanin as we clearly state. However, as our results go on to show, the most probable explanation for the distribution of, concentration of, and detailed coordination chemistry of the residues we find here is that part of the integument chemistry is derived from precursor pheomelanin. Please see the edits listed above.

Perhaps an additional final comment is required here. Organic geochemistry typically analyzes an undifferentiated “soup” of organic material derived from multiple precursor organisms and does so with no hope of fine-scale spatial resolution. Hence the “biomarkers” that stand out in typical organic geochemistry are from those organisms that dominate the biomass: plants and microbes in particular. The important and compelling conceptual difference here is that we know the precursor organism and we have spatial resolution at the scale of a few microns, so we therefore have constraints on what the starting assemblage of biochemistry would have been and a framework for how that material would have been distributed. For example the phosphorous and calcium in the bone structures could have been introduced by geochemical fluids, but such a scenario is unlikely: the distributions, concentrations, and coordination chemistry of the Ca and P match the original bone structure of this vertebrate and most probably are residual from the original organism. One could not make the same argument about the Ca dispersed through the sediment and associated with calcite- that would be illogical. Tests for endogeneity of material within bone could then be applied which would include measuring concentrations, determining coordination chemistry, and resolving organic compounds which would be consistent with collagen degradation. This is exactly our approach with soft tissue residue. The fact that we can resolve the spatial distribution of heterocyclic S and its correlation with organically bonded Zn within the fossil and that this matches the distribution of heterocyclic S and its correlation with organically bonded Zn within extant pheomelanin-rich comparators is an extremely strong result. If these organosulfur-Zn compounds were dispersed randomly within the sedimentary matrix it would be much more difficult to determine their provenance. However by analyzing these organic-metal complexes *in situ* we are making progress with respect to paleometallomics. A useful comparison is that the dominant Cu organic complex in a number of exceptionally preserved leaf fossils is a Cu double malate complex- identical to the coordination of Cu within the parenchyma of deciduous trees (see Edwards et al., 2014; “Leaf metallome preserved over 50 million years,” *Metallomics* 6, 774-782, DOI:10.1039/c3mt00242j). These plant derived complexes are completely different from the Cu-porphyrin complexes we resolve associated within specific animal soft tissue residues derived from hair, feathers, eyes, and ink sacs which we interpret as eumelanin biomarkers. We previously showed for the first time that Zn organosulfur complexes resolve pheomelanin pigmentation in modern organisms. This study is the first to indicate that Zn organosulfur complexes are likely to be the key to resolving the second most abundant pigment in the animal kingdom in fossil soft tissue.

It is the high fidelity of the distribution, concentration, and coordination chemistry of these organic compounds preserved within the discrete biological integumentary structures that strongly indicates that these compounds are the residue of the dominant organic compounds that were present in life. Within the *Apodemus atavus* integument, the dominant compounds will almost certainly have been alpha keratin and melanin pigments. Keratin does not significantly bind metals and does not contain heterocyclic sulfur. Pheomelanin strongly binds Zn and contains heterocyclic S. Identifying both aspects typical of precursor pheomelanin biochemistry within the fossilized integument is strong evidence that pheomelanin pigment residue is present. The fact that we get identical positive results from two specimens of this species but negative results from other fossil species from the same lithology further confirms our conclusion.

We sincerely hope that the edited manuscript now makes this clear. We have done our best to respond to these criticisms and have endeavoured to make the manuscript as clear as possible without adding too much in terms of length.

Reviewers' Comments:

Reviewer #2:

Remarks to the Author:

Among other improvements the revised manuscript "Pheomelanin pigment remnants mapped..." submitted to Nature Communications now includes figures of the counter slabs of the two Apodemus specimens.

However, my major concern is still the reconstruction of melanin pigment patterning of the fossil Apodemus based on SRS-XRF data. With the knowledge of present-day Apodemus pigment patterning, one would infer the reconstructed patterning of the closely related fossil species. However, I'm not convinced if the suggested reconstruction would be possible only based on the elemental mapping data shown in Fig 1 and 2.

In both fossil specimens no tissue material and fur is preserved in the tail. This part of the fossil consists only of bones (as indicated by the blue color of the SRS-XRF images due to the element phosphorus). This is not a problem of uneven separation, but a problem of lack of material. With the lack of tissue material and fur obviously no statement regarding the pigmentation of the tissue and fur of these parts is possible. By contrast, Rietschel and Storch (1974) (ref. 27 of the revised manuscript) have depicted a further specimen of Apodemus atavus from Willershausen (SMF M 4627), not investigated in this work, that shows preservation of tissue material in the tail. For the feet of the mice there is a similar situation. The feet almost only consist of bones with almost no preservation of tissue and fur.

Furthermore, in the reconstruction of pigment patterning the belly of Apodemus atavus is light colored with no melanin pigment, corresponding to the color patterning of present-day relatives (countershading of the fossil Apodemus indeed is very likely). However, I would not come to this conclusion from the elemental mapping data shown in Fig. 1B.

It is obvious that in Fig. 1A the thickness of the preserved tissue material and fur is not the same in different parts of the fossil showing different layers (dark brown and light brown areas). The counter slab shows a more homogenous distribution of soft tissue. Apparently, some material of the main slab was lost during splitting of slab and counter slab. The uneven distribution of elements in Fig. 1B clearly reflects the uneven thickness and distribution of fossil material.

In summary, in my view the work provides interesting insights in the elemental composition of different layers of tissue material of a fossil mammal. However, I see no evidence that the melanin pigment patterning of the extinct species can be restored based on SRS-XRF data.

One minor point:

The title "...in fossils of extinct mammals" is a bit misleading, because it may suggest that several species of mammals were investigated, not a specific species.

Reviewer #4:

Remarks to the Author:

In this paper, Manning et al. present a new chemical imaging method for the identification of pheomelanin in fossils. I have considered the manuscript, the reviewer's comments and rebuttal. The work is comprehensive, detailed, and focussed, and follows on logically from the group's previous work on melanin pigments in extant and extinct plants and animals. The work fills an important gap in

our understanding of the fossil record of pigments; this, plus the use of cutting-edge cross-disciplinary techniques, renders the paper of broad interest; as such it is certainly suitable for publication in Nature Communications.

The authors respond to the comments of Reviewer 2 and 3 in considerable detail. Both of these reviewers raised concerns about the endogeneity of the Zn and S in the fossils. The key data relating to this issue are robust: the spectroscopic data and maps provided by the authors clearly demonstrate that these moieties are concentrated in the fossil soft tissues, and moreover, that the Zn is chemically bound to an organic form of sulfur.

Reviewer 2 also noted a 'major concern' regarding the pigmented distribution as reconstructed by the authors. To be honest, I don't consider this a major issue as the major scientific advance that the paper represents is our ability to identify traces of phaeomelanin in fossils, not the precise pattern configuration on the animal. There is inherent uncertainty built into any reconstructions of fossil colour, and I think the efforts of the authors to characterise the distribution of soft tissues on part and counterpart address this issue satisfactorily, with the exception of one minor point: rather than stating 'we can only conclude that the ears and lower jaw region...were probably not pigmented' the interpretation here should be more tentative as the authors cannot exclude the possibility that these body regions were pigmented; this should be stated explicitly.

Reviewer 3 has a number of issues with the manuscript, in particular relating to the issue of endogeneity and the impact of diagenesis on the preserved chemical signals. It seems that this reviewer is perhaps not particularly familiar with fossil material, as he/she takes a very strong stance regarding the impossibility of identifying pigments in fossils. This view may have been appropriate 10 years ago, but now there is an impressive body of literature providing multiple lines of chemical (and morphological) evidence for the widespread survival of melanin pigment in fossils.

The authors provide comprehensive responses to the reviewer's various comments relating to this issue, and I am satisfied with their responses. However I think the authors do need to acknowledge the possibility (however small) that the complexes identified are not original. In this light, the paper would benefit from the inclusion of a statement along the following lines: '... although it is possible that the specific Zn-S complexes identified in the fossils were generated through diagenesis, there is as yet no experimental or other evidence for such a process. Based on the available evidence, the most parsimonious explanation is that the Z-S complexes are derived from phaeomelanin.'

Reviewer 3 also raises concerns about the fact that the structure of melanin isn't fully characterized in modern material and that metals are being used as a proxy for the preservation of the pigment. To be perfectly frank, it isn't necessary to fully visualise the structure of a molecule in order to identify it in fossils – indeed, it is unreasonable to expect molecules to survive in an intact state in fossils. Such expectations belie an unfamiliarity with fossil diagenesis and with the fossil record of biomolecules. There is extensive literature on the identification of biomolecules in fossils; most fossil biomolecules are altered from their original state, often showing signs of transformation to compounds that are more stable over geological time periods (yet retaining core components or moieties that allow identification of original precursor compounds *in vivo*). Based on the nature of the techniques used to analyse fossil material (usually based on the identification of molecular fragments or chemical bonds), it is sufficient to characterise the major molecular building blocks present. These arguments are also made by the authors in their response to Reviewer 3's comments.

The key point raised in this paper is that in certain, specific, cases, we don't need to identify molecular fragments – key metal-organic complexes can also be diagnostic of certain biomolecules. The sulfur present in the fossil is almost certainly organic, and is associated with Zn in an identical relationship to

the Zn-organosulfur moiety in extant phaeomelanin.

Further, as the authors correctly assert, any diagenetic alteration that does take place does not necessarily affect all components, so although benzo-S moieties may be diagenetically altered, it is not implausible that the core Zn-S complex may survive, particularly as other organometal complexes in melanin (relating to Cu in eumelanin) can survive fossilization.

In sum, I have experience with the XAS techniques used by Manning et al. and their analyses are robust and their arguments logical and cogent. This paper contrasts with some recent papers on fossil melanin that have over-emphasised colour patterns, which are notoriously qualitative, subjective, and inbuilt with various limitations. This paper provides rigorous chemical analyses by the leaders in the field, demonstrating the way that fossil colour research should be done. I strongly support publication.

Reviewer #5:

Remarks to the Author:

The paper presents mainly spectroscopic results related to the type of melanin present in the skin/fur of an extinct mammal (*A. atavus*). The analysis is supported by characterization of fresh mice fur and human hair of different color. Zn, due to its affinity with pheomelanin (red pigment), is used as a marker for the determination of the spatial distribution of this type of melanin that is related to the color of the extinct *A. atavus*. Although the bonding of Zn with pheomelanin (ref. 1) and SRS-XRF mapping of Zn in fossils [ref. 15] have been previously reported, in this work the spatial distribution of Zn is correlated with the presence of pheomelanin which is associated with the color of the fur. The correlation is not only revealed from the coexistence of Zn with S (which is present in this type of melanin) but also with XAFS spectra that provide chemical information. This is the novelty of the paper. The results are of interest for palaeontologists and biologists as they provide a way to obtain information on the color (and its spatial variation) throughout an organism for extinct species with an indirect manner.

The paper is well written providing adequate information on the samples and the experimental conditions. The samples are characterized by a considerable number of techniques. Discussion on the possible effects of fossilization and diagenesis is also included and along with the fossil samples, samples from modern animals and humans are analyzed for reference and comparison purposes. The paper can be accepted after minor revision.

My main concerns are the following:

* Page 4, L28-29, "... the bulk of the Zn imaged ... with organic S": The total area of green regions (Zn) in Fig. 1B is almost equal to the total yellow area. This observation does not justify the word "bulk" in the sentence.

* Page 4, L38-Page 5, L2, "Figure 2B ... Zn-cysteine": By tuning the energy of the incident beam to 2472.5 eV, not only the thiols but also the benzothiazole and the disulfide contributions are detected, because at this energy the spectra of these compounds are also characterized by considerable intensity (see Fig. 3). Selecting this energy just excludes the contribution of the sulfates which are present in the sediment.

*Page 5, L14-17: Which is the beam size during the acquisition of the S and Zn XAFS spectra? Please mention this value in the methods section and if possible indicate/mark the exact positions on the XRF maps shown in Figs 1 and 2 from where the XAFS spectra (*A. atavus* fur, fur2, scapular, lumbar) were recorded.

*Page 5, L27-28: Please provide the uncertainty in the determination of the 6% and 9% values of heterocyclic sulfur also in the main text. Using the uncertainty values provided in the supplementary material (Table S1), the benzothiazole percentage for the red mouse ranges from 0.07 to 0.11% and for the albino ranges from 0.05 to 0.07%. Therefore, the two concentrations are the same within the error bars. To my opinion the spectra and the analysis do not suggest that there is a statistically significant difference between the spectra of the red and the albino mice.

* Fig. 3: Are the spectra (extant and reference samples) normalized prior to LCA? If yes at which energy? Please mention that in the Figure caption.

* Fig. 3: Did you try to include in the LCA of the extant spectra also the spectrum of meth. sulfoxide? This one also has a strong peak close to the benzothiazole resonance.

* Fig. 3, fossil spectra: The contribution from the sediment dominates the spectra from the fossils and safe results on the S speciation percentages can not be safely obtained. Therefore the S spectra alone cannot unambiguously resolve pheomelanin residue, as it is correctly stated in Page 6, L17-20. Did you try to subtract the spectrum of the matrix from the spectra of the fossils and compare the new spectra (or try LCA) with the spectra of the fresh fur/hair?

* Caption of Fig. 4, L32: The features/shape of the XANES spectra are not determined solely from the atom that is bonded to Zn but rather from the bonding geometry/configuration around Zn. It is therefore more "correct" to mention the compound ZnS and ZnSO₄, or the Zn sulfide and sulfate instead of Zn-S and Zn-O.

* Fig. 4: Please provide a title for the vertical axis in 4A

* Table 1 (notes): So₂ should be converted to S₀². E₀ should be also converted to E₀. However, the energy shift is usually denoted as ΔE₀. Correct also the subscripts in ZnSO₄.6H₂O

* Page 6, L41: The summed coordination numbers in the 1st shell of the two fossil spectra (according to Table 1) are 5.3-5.6 and 4.5.

* Page 7, L1-6: please correct the distances according to the values listed in Table 1.

* Table 2: Human hair is rich in trace and essential metals which are not necessarily bonded to melanin. Their content strongly depends on environmental and nutritional factors, as well as on diseases/disorders. A single sample of one, e.g. blond, individual is not necessarily representative of all blonds. Please comment on that and add a relevant reference [see for example doi: 10.1016/0048-9697(95)91020-4]

* Page 8, L36, "XRF analyses ...S1)": The supplementary Inf. Table S1 (p. 26) does not contain XRF results.

* Page 3, L7: define the ToF-SIMS acronym

* Page 18, L2&13: define the DLS acronym the first time it appears in the manuscript

12-Feb-2019

Point by point rebuttal to the reviewers.

Below we present the *reviewers' comments in italics* and our response in normal typeface. Where necessary, we also include direct references to the points in the manuscript that we have edited in order to deal with the reviewer's specific comment. Changes or additions to the original text are underlined.

Reviewer #1

No additional comments.

Reviewer #2

Previously, this reviewer questioned our interpretation of remnant pigment patterning based on the concentrations of organosulfur-Zn compounds due to the possibility of uneven separation of soft tissue between part and counterpart of these fossils:

"Generally, one part of fossils from Willershausen is preserved on a slab, while another part of the fossil remains on the counter slab."

We provided additional photographic documentation to show that uneven separation is not the case with either of the specimens analysed here. This reviewer acknowledges our inclusion of these images:

Among other improvements the revised manuscript "Pheomelanin pigment remnants mapped..." submitted to Nature Communications now includes figures of the counter slabs of the two Apodemus specimens.

Nonetheless, this reviewer returns to the possibility of differential preservation as an alternative explanation to the chemical distribution we see here.

However, my major concern is still the reconstruction of melanin pigment patterning of the fossil Apodemus based on SRS-XRF data. With the knowledge of present-day Apodemus pigment patterning, one would infer the reconstructed patterning of the closely related fossil species...(However) The uneven distribution of elements in Fig. 1B clearly reflects the uneven thickness and distribution of fossil material.

In summary, in my view the work provides interesting insights in the elemental composition of different layers of tissue material of a fossil mammal. However, I see no evidence that the melanin pigment patterning of the extinct species can be restored based on SRS-XRF data.

This is the main criticism of this reviewer, and reviewer #4 also comments on this explicitly: “Reviewer 2 also noted a ‘major concern’ regarding the pigmented distribution as reconstructed by the authors. To be honest, I don’t consider this a major issue as the major scientific advance that the paper represents is our ability to identify traces of phaeomelanin in fossils, not the precise pattern configuration on the animal. There is inherent uncertainty built into any reconstructions of fossil colour, and I think the efforts of the authors to characterise the distribution of soft tissues on part and counterpart address this issue satisfactorily, with the exception of one minor point: rather than stating ‘we can only conclude that the ears and lower jaw region...were probably not pigmented’ the interpretation here should be more tentative as the authors cannot exclude the possibility that these body regions were pigmented; this should be stated explicitly.”

In order to address the comments of reviewers #2 and #4 with respect to this point we have edited the penultimate paragraph of the manuscript. We make it clear that this is a postulate and not a definitive result concerning patternation, despite the fact that there are two lines of evidence (our imaging and the extant phylogenetic bracket) which lend support to the postulated pigment distribution. However, in light of the guidance of the reviewers, we have removed the suggestion that the dorsal surface could have been more highly pigmented than the ventral surface and now only refer to the extremities vs. the corpus. We furthermore now explicitly state that differential preservation may complicate the interpretation of pigment distribution, see page 11, lines 37-41 and page 12 lines 1-9:

“The uneven distribution of Zn, Cu, and organic S suggests that the integument may not have been uniformly melanized. There is a lack of organically bound Zn in the extremities (i.e. tail, feet) of both *A. atavus* specimens as shown in the XRF images. Optical photographs show clearly that this is not due to preferential separation between part and counterpart, because the optical images show comparable quantities of soft tissue residue on each side of the bedding plane fossils (see Supplementary Figure 12). The lower quantities of organosulfur-Zn complexes in these regions compared to the corpus could reasonably be caused by lower *in vivo* quantities of pigment in these parts of the organism. Given that our data are consistent with regions of low pigment densities in analogous parts of extant related species (e.g. *A. flavicollis*) we postulate that these areas of *Apodemus atavus* could also have been weakly pigmented. However, we also note that relative to a third specimen of *Apodemus atavus*²⁷, soft tissue preservation in the tail regions of both of the specimens imaged here is less complete, and therefore differential preservation of soft tissue may be acting to complicate the assignment of original pigment patternation.”

The correlation maps and related spectroscopy from both fossil specimens however do indicate with a high degree of certainty that the corpus of this species was rich in both melanin pigments as in *A. sylvaticus*...”

We have also edited the last sentence in the abstract such that we do not mention patterning, page 1, lines 30-34:

“In this work we develop the chemical imaging methodology to show that organosulfur-Zn complexes are indicators of pheomelanin (red pigment) in extant and fossil soft tissue and that the mapping of these residual biochemical compounds can be used to restore melanin pigment distribution in a 3 million year old extinct species of mammal (*Apodemus atavus*).”

One minor point: The title “...in fossils of extinct mammals” is a bit misleading, because it may suggest that several species of mammals were investigated, not a specific species.

We have had five reviews of this manuscript and this is the first time anyone has questioned the title, however we would be willing to change the title to: “Pheomelanin pigment remnants mapped in extinct mammal fossils.”

Reviewer #3

No additional comments.

Reviewer #4

Besides commenting on the patternation criticism of reviewer #2 and suggesting a logical way to address that comment (which we deal with above), this reviewer provides a useful discussion of the review provided by reviewer #3 and our rebuttal to those comments. We appreciate the efforts of this reviewer to help put this research into context and we believe that this reviewer explains how our changes to the manuscript in order to deal with the first round of comments by reviewer #3 adequately address those comments. This reviewer raises no further points.

Reviewer #5

This new reviewer recommends only minor revision and is positive about the findings and the supporting data. This reviewer clearly has spectroscopic experience and has highlighted a few technical aspects which could be revised to improve clarity. We address these below on a point by point basis.

** Page 4, L28-29, "... the bulk of the Zn imaged ... with organic S": The total area of green regions (Zn) in Fig. 1B is almost equal to the total yellow area. This observation does not justify the word "bulk" in the sentence.*

We did provide statistical and imaging information concerning the correlation of organic sulfur with Zn which indeed supports our statement that the bulk of the Zn imaged is "bound within the integument and is associated with organic S." We referred to the correlation maps presented as Supplementary Figure 2 in the previous sentence, page 4 lines 29-30: "Supplementary Figure S2 presents computed correlation maps of Zn with organic S and Zn with Cu in both of the *A. atavus* specimens used in this study." However, the fact that the reviewer makes this comment means that we did not present our data clearly enough. In order to improve clarity in this regard, we have edited the text to explicitly refer to Supplementary Figure 2 with respect to this assertion, page 4, lines 30-32:

"The clear correlation of zinc with organic sulfur highlighted in Figure 1B and Supplementary Figure 2 robustly shows that the bulk of the Zn imaged is bound within the integument and is associated with organic S."

** Page 4, L38-Page 5, L2, "Figure 2B ... Zn-cysteine": By tuning the energy of the incident beam to 2472.5 eV, not only the thiols but also the benzothiazole and the disulfide contributions are detected, because at this energy the spectra of these compounds are also characterized by considerable intensity (see Fig. 3). Selecting this energy just excludes the contribution of the sulfates which are present in the sediment.*

Tuning to a specific resonance highlights the contribution of a single oxidation state to the XRF image, but the reviewer is correct in commenting that statistically there is a contribution from neighbouring oxidation states, especially when the resonances are close in energy. We do not say anything which might mislead the reader in this regard within the main text, although we agree that the Figure caption could be clearer. We have edited the figure caption for Figure 1 to read:

"...plus a third map which has been produced to especially emphasize the distribution of a specific oxidation state of organic sulfur (red = S in thiol) in order to highlight the clear correlation between the distribution of Zn and organic sulfur which appears as bright yellow."

**Page 5, L14-17: Which is the beam size during the acquisition of the S and Zn XAFS spectra? Please mention this value in the methods section and if possible indicate/mark the exact positions on the XRF maps shown in Figs 1 and 2 from where the XAFS spectra (A. atavus fur, fur2, scapular, lumbar) were recorded.*

Approximate beam sizes have been added for the S XANES on page 13, lines 34-35; "using a defocussed beam with slit apertures of 500 μ m (vertical) and 50 μ m (horizontal)", and for the Zn EXAFS on page 13, line 40: "slit aperture same as above".

We have also added annotation to Supplementary Figure 8 in order to indicate approximately where the Zn K-edge EXAFS were measured:

"Supplementary Figure 8 (continued). B) Point locations for heavy element SRS-XRF analyses (filled circles) and Zn K-edge EXAFS analyses (open circles) of the lateral fossil indicated on the zinc map, with concentration results table keyed to the map. L0 refers to the surface organic film, L1 refers to the underlying rock matrix. (Point analyses from SSRL, beamline 6-2; EXAFS from DLS, beamline I18.)

**Page 5, L27-28: Please provide the uncertainty in the determination of the 6% and 9% values of heterocyclic sulfur also in the main text. Using the uncertainty values provided in the supplementary material (Table S1), the benzothiazole percentage for the red mouse ranges from 0.07 to 0.11% and for the albino ranges from 0.05 to 0.07%. Therefore, the two concentrations are the same within the error bars. To my opinion the spectra and the analysis do not suggest that there is a statistically significant difference between the spectra of the red and the albino mice.*

We have edited the text such that we do not claim “higher” quantities of benzothiazole in the red fur, page 5, lines 31-35:

“A binary linear combination fit of benzothiazole plus disulfide for each extant specimen shows that the red fur contains ~9% (± 2) heterocyclic sulfur and the albino fur contains ~6% (± 2) heterocyclic sulfur. Organosulfur compounds have been shown to be a key component of pheomelanized tissue and to bond directly to divalent metals²⁹. Therefore the heterocyclic sulfur in the red fur is almost certainly due to the presence of organosulfur compounds associated with pheomelanin pigment.”

** Fig. 3: Are the spectra (extant and reference samples) normalized prior to LCA? If yes at which energy? Please mention that in the Figure caption.*

Yes, the spectra are normalized. We have noted this in the caption for Figure 3 as requested:

“Normalized spectra from the fossil are presented along with LCF fits...”

In the interest of brevity we have not put further detail about the LCF method in the main manuscript. Instead, we have edited the title of Supplementary Table 1 and added a footnote to the table.

“Table S1. Sulfur speciation from normalized* XANES LCF analyses...”

“*LCF normalization was accomplished by setting the edge jump between the pre-edge and post edge to be equal to 1. Pre-edge was typically -20 to -10 eV relative to the critical energy. The post-edge normalization region varied depending on the analyte.

Typically the post-edge was taken as 8 to 30 eV above the critical energy, extending between 15 to 150 eV above the critical energy. Beam induced photo-oxidation is a problem with sulfur speciation analysis using a synchrotron beam, and we performed many preliminary scans in order to optimize scan times and scan ranges such that the XANES spectra were not affected by oxidation but were wide enough in energy to allow normalization above the edge for all of the specimens analyzed.”

** Fig. 3: Did you try to include in the LCA of the extant spectra also the spectrum of meth. sulfoxide? This one also has a strong peak close to the benzothiazole resonance.*

No, as we state in the manuscript this was a two component fit. Because the extant material is all unaged we did not think it was chemically valid to include an oxidized component. This would most likely improve the fit, but only because it would include another degree of freedom. If we had been dealing with archaeological standards or standards subjected to high temperature that would have been chemically justified, but given the pristine nature of our extant standards this did not seem appropriate.

** Fig. 3, fossil spectra: The contribution from the sediment dominates the spectra from the fossils and safe results on the S speciation percentages can not be safely obtained. Therefore the S spectra alone cannot unambiguously resolve pheomelanin residue, as it is correctly stated in Page 6, L17-20. Did you try to subtract the spectrum of the matrix from the spectra of the fossils and compare the new spectra (or try LCA) with the spectra of the fresh fur/hair?*

We explored the data in several ways that are not included in this manuscript, however we did not want to present “matrix subtracted” spectra, because some of the sulfate within the fossil “fur” areas may actually be derived from the mouse itself. We take the reviewer’s point, but we feel the Figure as shown makes it clear that the fossil preserves organic sulfur species comparable to those present in extant hair and fur, and we feel that the spectra do not need further processing in that regard.

** Caption of Fig. 4, L32: The features/shape of the XANES spectra are not determined solely from the atom that is bonded to Zn but rather from the bonding geometry/configuration around Zn. It is therefore more “correct” to mention the compound ZnS and ZnSO₄, or the Zn sulfide and sulfate instead of Zn-S and Zn-O.*

The reviewer is correct, however the first coordination shell strongly dominates XANES spectra, and so for clarity we have changed the caption for Figure 4 so that there is no

confusion: “The two vertical lines indicate the absorption spectrum maxima for pure first shell Zn-S and pure first shell Zn-O species.”

* Fig. 4: Please provide a title for the vertical axis in 4A

Apologies, we have added the title “Amplitude.”

* Table 1 (notes): So_2 should be converted to SO_2 . E_o should be also converted to E_0 . However, the energy shift is usually denoted as ΔE_0 . Correct also the subscripts in $ZnSO_4 \cdot 6H_2O$

All three formatting typos have been fixed. We have also added units to the radial distance measurement.

* Page 6, L41: The summed coordination numbers in the 1st shell of the two fossil spectra (according to Table 1) are 5.3-5.6 and 4.5.

We thank the reviewer for spotting this, it is an important typo. This has been corrected to read, page 7, lines 3-5:

“The summed coordination numbers for the first shell in the two fossil spectra are 5.6 and 5.3...”

* Page 7, L1-6: please correct the distances according to the values listed in Table 1.

Again we thank the reviewer, the text was referring to an earlier version of the table. The text and table are now consistent, page 7, lines 5-9:

“The O1 and O2 bond distances are consistent with these two coordination states, where the shorter oxygen bond lengths (i.e. 1.983 and 1.993 Å) imply four-fold oxygen coordination (ideal distance of 1.96 Å) and the longer bonds (i.e. 2.104 and 2.056 Å) are comparable to the range measured for Zn in six-fold oxygen coordination (2.06-2.184 Å). The sulfur in the first shell of both the fossil and extant fur/hair has bond distances (2.266-2.352 Å)...”

* Table 2: Human hair is rich in trace and essential metals which are not necessarily bonded to melanin. Their content strongly depends on environmental and nutritional factors, as well as on diseases/disorders. A single sample of one, e.g. blond, individual is not necessarily

representative of all blonds. Please comment on that and add a relevant reference [see for example doi: 10.1016/0048-9697(95)91020-4.]

Keratinous integument indeed contains variable quantities of a range of trace metals. There are many environmental studies of trace metals in keratinous integument, indeed we have participated in studies of elevated As concentrations in fingernails and hair from southeast Asia correlated with contamination levels in drinking water (see: Gault A.G., Rowland H.A.L., Charnock J.M., Wogelius R.A., Gomez-Morilla I., Vong S., Leng M., Samreth S., Sampson M.L., Polya D.A. (2008) "Arsenic in hair and nails of individuals exposed to arsenic-rich groundwaters in Kandal Province, Cambodia," *Science of the Total Environment* **393**, 168-176). However, while concentration data concerning extant integument is relatively abundant, there is much less information concerning the spatial distribution of or coordination chemistry of trace metals in integument. This research group published the first detailed study of Cu in eumelanised tissue (reference #2 in our manuscript) and the first detailed study of Zn in both eumelanized and pheomelanized tissues (reference #1 in our manuscript). We never suggested that the data of Table 2 represent a statistical sampling of all pheomelanized tissues, the point here is that the concentrations of Zn and Cu within the fossil fur and the ratios of Zn/Cu are comparable to the seven analyses determined using our methods that we present of extant hair and fur (as opposed to feathers or fingernails) covering a range of pigment densities. This strongly supports the endogeneity of these trace metals and allows us to use the coordination chemistry to resolve information about the original organism. In fact, upon reflection, this reviewer has reminded us of one of the key motivations for doing this type of work; point analyses or bulk analyses are almost useless in understanding pigmentation unless they are completed along with imaging and spectroscopy. This is the fundamental reason why we are pursuing this line of scientific inquiry, and the reviewer's request for us to address the trace metal variability in soft tissue when introducing the data on Table 1 and 2 is helpful. We have added the following short passage in order to respond, page 8, lines 15-20:

"Trace metal loading within keratinous integument is complex. Extant specimens for comparison to the fossil data were chosen as representatives of different pigmentation types and are meant to show relative concentrations. The well-known variability in metal concentrations in feathers, hair, fingernails, and other tissues is precisely why elemental mapping is so crucial and why bulk metal analysis can be misleading with respect to integument. For these reasons we also present ratios of Zn/Cu in Table 2 in an attempt to account for the range in concentrations observed in extant and fossil tissue."

** Page 8, L36, "XRF analyses ...S1)": The supplementary Inf. Table S1 (p. 26) does not contain XRF results.*

Actually, Supplementary Table 1 does contain XRF data along with the LCF analytical results, but we understand the confusion. We have edited the Table title to clarify this.

“Table S1. Sulfur speciation from normalized* XANES LCF analyses and total sulfur XRF quantification (Sulfur quantification 2σ error = $\pm 0.4\%$ absolute). All data from DLS I18.”

** Page 3, L7: define the ToF-SIMS acronym*

Page 3, lines 9-10: “Time of Flight –Secondary Ion Mass Spectrometry (ToF-SIMS)...”

** Page 18, L2&13: define the DLS acronym the first time it appears in the manuscript.*

Page 13, line 12: “The microfocus beam line I18 at the Diamond Light Source (DLS)...”